



i$_N$RACM: Incorporating $^{15}$N into the Regional Atmospheric Chemistry Mechanism (RACM) for
assessing the role photochemistry plays in controlling the isotopic composition of NO$_x$, NO$_y$, and
atmospheric nitrate.

Greg Michalsk[1], Huan Fang[1], Wendell W.Walters[2], and David Mase[1].
*[1]Purdue University, Department of Earth, Atmospheric, and Planetary Sciences, Department of*
*Chemistry. West Lafayette, IN. USA*
*[2]Brown University, Institute for Environment and Society. Providence, RI. USA*
Correspondence: Greg Michalski, gmichals@purdue.edu
Key Points
• Modeling nitrogen isotope fractionation during the photochemical oxidation of nitrogen
oxides into atmospheric nitrate.
• Incorporation of N isotopes of NOy into the Regional Atmospheric Chemistry
Mechanism.
• Implications for quantifying NO$_x$ sources and oxidation pathways using nitrogen
isotopes.





Abstract
Nitrogen oxides, classified as $NO_x$ (nitric oxide (NO) + nitrogen dioxide ($NO_2$)) and $NO_y$ ($NO_x$ +
$NO_3$, $N_2O_5$ $HNO_3$, + $HNO_4$ + HONO + Peroxyacetyl nitrate (PAN) + organic nitrates + any
oxidized N compound), are important trace gases in the troposphere, which play an important
role in the formation of ozone, particulate matter (PM), and secondary organic aerosols (SOA).
Among many uncertainties in movement of atmospheric N compounds, nowadays understanding
of $NO_y$ cycling is limited by $NO_x$ emission budget, unresolved issues within the heterogeneous
uptake coefficients of $N_2O_5$, the formation of organic nitrates in urban forests, etc. A
photochemical mechanism used to simulate tropospheric photochemistry was altered to include
$^{15}N$ compounds and reactions as a means to simulate $\delta^{15}N$ values in $NO_y$ compounds. The 16 N
compounds and 96 reactions involving N used in Regional Atmospheric Chemistry Mechanism
(RACM) were replicated using $^{15}N$ in a new mechanism called $i_N$RACM. The 192 N reactions in
$i_N$RACM were tested to see if isotope effects were relevant with respect to significantly changing
the $\delta^{15}N$ values (+/- 1‰) of $NO_x$, HONO, and/or $HNO_3$. The isotope fractionation factors ($\alpha$) for
relevant reactions were assigned based on recent experimental or calculated values. Each
relevant reaction in the $i_N$RACM mechanism was tested individually and in concert in order to
assess the controlling reactions. The final mechanism was tested by running simulations under
different conditions that are typical of pristine, rural, urban, and highly polluted environments.
The results of these simulations predicted several interesting $\delta^{15}N$ variations.



## 1. Introduction

Nitrogen oxides are an integral part of atmospheric chemistry, controlling the oxidation state of the troposphere, influencing aerosol formation, altering the pH of rainwater, and facilitating the movement of nitrogen through the N cycle. Nitrogen oxides are classified as $NO_x$ (nitric oxide (NO) + nitrogen dioxide ($NO_2$)) and $NO_y$ ($NO_x$ + $NO_3$, $N_2O_5$ $HNO_3$, + $HNO_4$ + HONO + Peroxyacetyl nitrate (PAN) + organic nitrates + any oxidized N compound) [*Day et al.*, 2003; *Harrison et al.*, 1999; *Hegglin et al.*, 2006; *Ma et al.*, 2013]. $NO_x$ produces ozone ($O_3$) through $NO_2$ photolysis, and NO acts as a catalyst in $O_3$ production when volatile organic compounds (VOCs) are present. In turn, $O_3$ photolysis generates OH radicals, which initiates a radical chain reaction involving $HO_2$ and organic peroxide propagators that result in the oxidation of chemically reduced compounds in the troposphere making them more soluble [*Finlayson-Pitts and Pitts*, 2000; *Seinfeld and Pandis*, 1998]. Thus, $NO_x$ facilitates the cleansing of the atmosphere through the production of $O_3$ and $OH_x$ (OH + $HO_2$), which together define the troposphere's oxidation state [*Bloss et al.*, 2005; *Lelieveld et al.*, 2008; *Prinn*, 2003]. These oxidants play an important role in the formation of particulate matter (PM) [*Bauer et al.*, 2007; *Pye et al.*, 2010], forming secondary organic aerosols (SOA) via VOC oxidation [*Hoyle et al.*, 2011; *Shrivastava et al.*, 2017]. They also generate secondary inorganic PM through $NO_x$, sulfur oxides ($SO_x$), and ammonia ($NH_3$) neutralization, which leads to ammonium nitrate ($NH_4NO_3$) and ammonium sulfate (($NH_4$)$_2SO_4$) production [*Cao et al.*, 2017; *Pan et al.*, 2018; *Pusede et al.*, 2016]. The production of PM has important consequences for air quality aerosols [*Andreae and Crutzen*, 1997], human health [*Bruningfann and Kaneene*, 1993; *Hall et al.*, 1992], and radiative forcing [*Charlson et al.*, 1992; *Chen et al.*, 2007]. Termination reactions in $NO_y$ cycling produces $HNO_3$, and facilitates the production of sulfuric acid ($H_2SO_4$), two strong acids that decrease the pH of rain, known colloquially as acid rain and impact aerosol pH, both of which triggers a number of negative impacts on the environment [*Brimblecombe et al.*, 2007; *Lajtha and Jones*, 2013]. When $NO_y$ is deposited to the surface by wet and dry deposition, it transfers bioavailable N to ecosystems that may be bereft of, or saturated with, bioavailable N. This process can shift the balance of both terrestrial and aquatic ecosystems and impact the goods and services that those ecosystems can normally deliver [*Du et al.*, 2019; *E. M. Elliott et al.*, 2019; *Fowler et al.*, 2013]. Thus, understanding $NO_y$ sources and their chemistry is important for an array of scientific disciplines and evaluating their social, economic, and cultural impact on the environment.

Despite this importance, there are numerous knowledge gaps in the understanding of the cycling of $NO_y$ in the atmosphere. The $NO_x$ emission budget is still poorly constrained. Most emission inventories rely on fixed emission factors for some sources that may, in fact, be variable. For example, power plant $NO_x$ emissions are based on assumed efficiency of catalytic converters that may not be accurate [*Srivastava et al., 2005; Felix et al., 2012*]. Soil NO emissions are highly dependent on soils moisture, redox conditions, fertilizer application rates, type, and timing making them challenging to constrain [*Shepherd, 1991; Galloway et al., 2004; Hudman et al., 2012; Houlton et al., 2013; Pilegaard, 2013*]. There are several unresolved issues with the heterogeneous uptake coefficients of $N_2O_5$ [*Brown et al.*, 2001; *Brown et al.*, 2006; *Chang et al.*, 2011] and the formation of organic nitrates in urban forests [*Lee et al., 2016; Romer et al., 2016; Kastler and Ballschmiter*, 1998]. The relative importance and mechanism of HONO formation versus HONO emissions are also hotly debated. Likewise, reactions of $NO_y$ in



the aqueous phase and mixed aerosols are not fully understood. Chemical transport models
(CTMs) do not accurately predict aerosol nitrate concentrations or other $NO_y$ mixing ratios [*Spak*
*and Holloway*, 2009; *Zhang et al.*, 2009]. Therefore, it is important that these uncertainties in
$NO_y$ cycling be resolved if we aim to have accurate air quality forecast and accurate chemistry-
climate models that use CTMs.
It has been suggested that stable N isotopes can provide clues as to the origin of $NO_x$
[*Elliott et al.*, 2009; *Felix and Elliott*, 2014; *Walters et al.*, 2015b] and the oxidation pathways
that transform in $NO_y$ [*Walters and Michalski*, 2015; 2016]. Isotopic measurements of $NO_y$
compounds show a wide range of $\delta^{15}N$ values (Eq. (1)), which has been suggested to indicate
variability in $NO_x$ emission sources, chemical processing, and/or a combination of these effects.
$\delta^{15}N$ is defined by the relative difference between the $^{15}N/^{14}N$ ratio in a $NO_y$ compound and the
ratio in air $N_2$ (the arbitrary reference compound) and is typically reported in parts per thousand
e.g. per mil (‰)
$\delta^{15}N_{NOy}$ (‰) = [($^{15}NO_y/^{14}NO_y$) / ($^{15}N_2/^{14}N_2$) -1]*1000         Eq. (1)
A number of studies have measured the $\delta^{15}N$ values of $NO_x$ collected from $NO_x$ sources such as
power plants [*Felix et al.*, 2012], automobiles [*Walters et al.*, 2015a], biomass burning [*Fibiger*
*and Hastings*, 2016], and non-road sources [*Felix and Elliott*, 2014].
Many studies have measured the $\delta^{15}N$ values of various $NO_y$ compounds collected from
the troposphere. Most of the $NO_y$ $\delta^{15}N$ data is for nitrate that has been collected on filters ($PM_{2.5}$,
$PM_{10}$, TSP) [*Moore, 1977; Savard et al., 2017*], as the dissolved $NO_3^-$ anion in rain [*Heaton,*
*1987; Hastings et al., 2003; Felix et al., 2015; Yu & Elliott, 2017*], or as gas phase $HNO_3$ [*Elliott*
*et al., 2009; Savard et al., 2017*]. The range of tropospheric $NO_y$ $\delta^{15}N$ values span -50 to +15‰
but the average is ~0‰. Two hypotheses have been offered to explain these ranges:  Source and
Photochemistry.  The source hypothesis [*Elliott et al.*, *2007; Hastings et al., 2013*] suggesting
that the tropospheric $NO_y$ $\delta^{15}N$ value range reflects the spatial and temporal mixing of $NO_x$
sources with different $\delta^{15}N$ values that is then converted into $NO_y$. The photochemistry
hypothesis [*Freyer*, 1978; *Freyer et al.*, 1993; *Walters et al.*, 2018] suggests that the observed
$NO_y$ $\delta^{15}N$ variations arise via isotope effects occurring when photochemical cycling partitions N
into the myriad of $NO_y$ compounds. These two hypotheses are not mutually exclusive.  Indeed it
is likely to be a combination of both processes, but their relative importance likely shifts
depending on environmental conditions such as a region's $NO_x$ source diversity, plume versus
dispersed chemistry, photolysis intensity, and oxidant load.  In turn, the $\delta^{15}N$ data might be a
new key to reconciling some of the current uncertainties in $NO_y$ sources and chemistry, if it can
be properly interpreted.
What has become clear is that the temporal-spatial heterogeneity of $NO_x$ sources and the
complex photochemistry of $NO_y$ presents a serious challenge to deciphering the observed $NO_y$
$\delta^{15}N$ values.  Except for a few isolated cases, a proper assessment of $NO_y$ $\delta^{15}N$ values will
require incorporating isotope effects into 3-D chemical transport models.  This will include
emission modeling of $^{15}NO_x$, meteorological mixing, factoring in isotope effects during $NO_y$
removal processes, and developing chemical mechanisms that incorporate $^{15}N$ compounds and
their relative rate constants. Here we take the first step in this endeavor by developing a chemical
mechanism (0-D photochemical box model) that explicitly includes $^{15}NO_y$ compounds and the
isotope effects that occur during their cycling through photolysis, equilibrium, and kinetic
reactions.



## 2. Methods

### 2.1 Chemical and isotopic compounds and reactions included in $i_N$RACM

The basis of the $i_N$RACM model is incorporating $^{15}N$ into the Regional Atmospheric Chemistry Mechanism (RACM) detailed in Stockwell et al. [*Stockwell et al.*, 1997]. RACM is an extension of the chemical mechanism used in the Regional Acid Deposition Model (RADM2) [*Stockwell et al.*, 1990], but with updated inorganic and organic chemistry. There are 17 stable inorganic compounds, 4 inorganic intermediates, 32 stable organic compounds, including 4 biogenic organics, and 24 organic intermediates in the RACM mechanism. The RACM mechanism uses 237 chemical reactions, including 23 photolysis reactions [*Atkinson, 1990; Atkinson et al., 1992*]. The rate constants, photolysis cross-sections and quantum yields for the inorganic compounds were taken from [*DeMore et al.*, 1994]. The RACM mechanism aggregates the thousands of VOC in the atmosphere into 16 anthropogenic and 3 biogenic organic compounds. Part of the aggregation criteria is based on the reactivity of a VOC towards the hydroxyl radical (•OH). Full details on how •OH reacts with alkanes, alkenes, aromatics, and other VOCs, and the aggregation rationale, can be found in Stockwell et al. (*1997*). The actinic flux model used in RACM was developed by Madronich (*1987*) and calculates the wavelength-dependent photon flux as a function of solar zenith angle, which is a function of time (hourly), date, latitude, and longitude. Inputs to the model include temperature, water vapor content, pressure, initial gas mixing ratios and primary pollutant emission rates. Complete details on the RACM mechanism can be found in Stockwell et al. (1997). The numerical solver used was VODE, part of the ODEPACK, a commonly used, and validated, collection of initial value ordinary differential equation solvers [*Brown et al.*, 1989; *Hindmarsh*, 1983].

Our $i_N$RACM (isotope N in RACM) mechanism was generated by adding $^{15}N$ isotopologues for the 2 primary (NO, $NO_2$) and the 11 secondary N pollutants found in the original RACM mechanism (Table S1a). By definition, an isotopologue is a compound with the same chemical formula but a different mass (e.g. NO = 30 amu and $^{15}NO$ = 31 amu, with N = $^{14}N$). This is different from isotopomers, which are isotopic isomers, compounds with the same mass but a different structure caused by isotopic substitution (e.g. $^{15}NNO_5$ = 109 amu and $N^{15}NO_5$ = 109 amu). Of all the reactive N compounds only $N_2O_5$ has multiple possible $^{15}N$ substitutions and 2 isotopologues were defined in the $i_N$RACM: $^{15}NNO_5$ and $^{15}N^{15}NO_5$. The $^{15}N$ compounds are numbered (Table S1a) and subscripted (a, b) in order to maintain a compound numbering scheme that is consistent with that in Stockwell et al. (*1997*). The non-N compounds found in both RACM and $i_N$RACM mechanisms have been excluded from Table S1a for the sake of brevity but can be found in Stockwell et al. (*1997*). The 16 $^{15}N$ compounds (Table S1a) were added to the original RACM FORTRAN code provided by Stockwell by using $Z =^{15}N$ (e.g. $^{15}NO$ is defined as ZO).

The 96 chemical reactions involving N compounds (Table S2a-f) were inspected and replicated for $^{15}N$ based on classification as the reaction being either "N only" or "multiple N" reactions. Single N reactions are those where only one N compound was found in the products and reactants, for example $NO + O_3 \rightarrow NO_2 + O_2$. Multiple N reactions could have multiple N compounds in the reactants, the products, or both. Examples of these possible multiple N reactions are $NO_2 + NO_3 \rightarrow N_2O_5$, $N_2O_5 \rightarrow NO_2 + NO_3$, and $NO_3 + NO \rightarrow NO_2 + NO_2$ respectively. For these multiple N reactions, a reaction probability was factored into the isotopologue/isotopomer rate constants (discussed below). For example, the N





isotopologue/isotopomer equivalent of the $N_2O_5 \rightarrow NO_2 + NO_3$ reaction has two isotopomer
reactions: $^{15}NNO_5 \rightarrow ^{15}NO_2 + NO_3$ and $^{15}NNO_5 \rightarrow NO_2 + ^{15}NO_3$. These two isotopologue rate
constants (R54a, R54b) are multiplied by a factor of 1/2 to account for this statistical probability.
Similar statistical factors were considered when N compounds or intermediates decomposed or
reacted to form multiple N products (R52a, R52b, R52c, R52d). All N isotopologue reaction
stoichiometry is given in Table S2a-f.
## 2.2 Isotope effects included in $i_N$RACM
The main challenge for developing realistic isotopologue chemistry in $i_N$RACM is
quantifying the differences in rate constants caused by isotopic substitution. These isotope
effects can be classified into four general types: Equilibrium isotope effects (EIE), kinetic
isotope effects (KIE), photo-induced isotope fractionation effects (PHIFE), and vapor pressure
isotope effects (VPIE). For this study, the most up-to-date isotope fractionations were used
when establishing the framework for modeling their effect associated with $NO_x$ oxidation
chemistry. The established framework will easily enable an adjustment of isotope effects as we
improve our understanding of isotope fractionation factors.
Urey *(1947)* and Bigelesien and Mayer *(1947)* showed that EIEs are driven by the
sensitivity of molecular and condensed-phase vibrational frequencies to isotopic substitutions
[*Bigeleisen and Mayer*, *1947; Urey, 1947*]. Because vibrations are used in the molecular
partition function (Q) to calculate equilibrium constants, isotopic substitution results in
isotopologues having different equilibrium constants. Urey [*1947*] defined the reduced partition
function ratio for two isotopologues of the same compound as a β value. For example, the
reduced partition function ratio of nitric oxide N isotopologues is $Q_{15NO}/Q_{NO} = \beta_{NO}$, with the
heavy isotope placed in the numerator by convention. The ratio of two β values is denoted as
$\alpha_{\beta1/\beta2}$ the isotope fractionation factor. For example, $\alpha_{NO/NO2}$ is the temperature-dependent isotope
fractionation factor (EIE) for the NO + $^{15}NO_2 \leftrightarrow ^{15}NO + NO_2$. In this case, at 298K $\beta_{NO} =$
1.0669 and $\beta_{NO2} = 1.1064$ and $\alpha_{NO/NO2} = \beta_{NO}/\beta_{NO2} = 0.9643$ [*Walters and Michalski, 2015*].
A KIE is the relative change in the rate of a unidirectional chemical reaction when one of
the atoms of the reactants is substituted with an isotope [*Bigeleisen and Wolfsberg*, *1958*]. KIEs
are driven by the change in energy required to proceed over the reaction barrier (transition state)
as well as changes in the probability of quantum mechanical tunneling [*Wolfsberg* et al., *2010*].
This generally results in a lighter isotopologue reacting faster than a heavier isotopologue. Much
of the early research on KIEs were investigations of the KIE in reactions containing hydrogen
isotopes and these studies usually defined a KIE = $k_L/k_H = \alpha_{L/H}$, where the k's are the rate
constants for the light and heavy isotopologues. This is the inverse of the definition of α usually
used in research dealing with EIE, VPIE, PHIFE and this inversion can lead to confusion. In this
paper, in order to maintain consistency between the α values for EIE, KIE, VPIE, and PHIFE, α
will be defined as heavy/light for all four effects.
The α values for EIE and KIE can be determined using a number of approaches. The α
values for EIE can be calculated if molecular constants (e.g. harmonic frequencies and
anharmonicity constants) of the isotopologue pair are known. Accurate molecular constants are
difficult to accurately measure for large molecules and as a result, they primary exist only for
diatomic and triatomic isotopologues [*Richet et al.*, *1977*]. The only experimental EIE values for
$^{15}N$ isotopologues of $NO_y$ is for the EIE between NO and $NO_2$ [*Sharma et al., 1970; Walters et*
*al., 2016*]. To determine the EIE in other $NO_y$ compounds we must rely on quantum chemistry

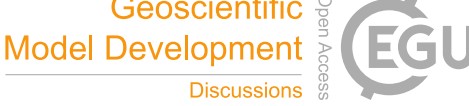

computation methods to estimate the molecular constants and anharmonicity constants. Recent
works utilizing these methods have estimated the EIE for most non-organic $NO_y$ compounds
[*Walters and Michalski*, 2015]. For KIE, in addition to molecular constants, the transition state
vibrational frequencies are also needed. The only $^{15}N$ KIE calculation to date for an $NO_y$
compound is for the KIE for the $NO + O_3$ reaction [*Walters and Michalski*, 2016].
These EIE and KIE values have been incorporated in $i_N$RACM in this study Table S2a-c.
If there is no isotope effect associated with any of the $NO_y$ reactions, then $\alpha$ is set equal to 1. The
$^{15}N$ isotopologue rate constant for any reaction is then $\alpha^{14}k$ where $^{14}k$ is the rate constant for any
$^{14}N$ reaction in RACM and these are given in Table S2a-f. It is useful to define the magnitude of
EIE and KIE in the same per mil (‰) notation used to quantify a $\delta^{15}N$ values by defining an
enrichment factor $\varepsilon(‰) = (\alpha-1)1000$. For example, the $NO_x$ isotope exchange equilibrium
mentioned above, the $\varepsilon_{NO/NO2} = -35.7‰$. This means that $^{15}NO/NO$ ratio would be 35.7‰ smaller
than the $^{15}NO_2/NO_2$ ratio if the isotopes in two gases were statistically distributed (Table S2b).
PHIFE is the relative change in photolysis rates of isotopologues due to the substitution
of a heavier isotope [*Yung and Miller, 1997*]. In the atmospheric N cycle, $NO_2$, $NO_3$, $N_2O_5$, and
HONO readily undergo photolysis at wavelengths of light that penetrate into the troposphere.
The PHIFE can be estimated using a simple zero-point energy shift model ($\Delta$ZPE). In this
approximation, the absorption spectra of the heavier isotopologue is generated by applying a
uniform blue shift (equal to $\Delta$ZPE) to the measured spectral absorbance of the light (major)
isotopologue [*Blake et al.*, 2003; *Liang et al., 2004; Miller and Yung, 2000*]. This results in
isotopic fractionation because the wavelength ($\lambda$) dependent photolysis rate constant ($J(\lambda)$) is
dependent on the convolution of the absorption cross-section ($\sigma(\lambda)$), actinic flux ($F(\lambda)$), and
quantum yield ($\phi(\lambda)$) (Eq. (2)):
$$^xJ(\lambda) = {}^x\sigma(\lambda)F(\lambda)\phi(\lambda) \quad \text{Eq. (2)}$$
The overall photolysis rate constant ($^zJ$) can be calculated by integrating $\sigma$, $F$, and $\phi$ over a range
of wavelengths that can cause dissociation ($\lambda_1$ and $\lambda_2$):
$$^xJ = \int_{\lambda_1}^{\lambda_2} {}^x\sigma(\lambda)F(\lambda)\phi(\lambda)d\lambda \quad \text{Eq. (3)}$$
The N isotopologue fractionation ($\alpha$) resulting from photolysis (of $NO_2$ isotopologues) is
calculated by (Eq. (4)).
$$\alpha_{47/46} = \frac{^{47}J}{^{46}J} \quad \text{Eq. (4)}$$
It is important to note that there are limitations in the $\Delta$ZPE-shift model [*Blake et al., 2003;*
*Liang et al., 2004; Miller and Yung, 2000*]. These include the failure to account for changes in
shape and intensity of absorption spectra upon isotopic substitution and the same quantum yield
(as a function of wavelength) is assumed for all isotopologues. Despite these limitations, this
approach should still give a rough estimate of photolytic fractionation until experimentally
determined PHIFE's become available [*Blake et al.*, 2003; *Liang et al.*, 2004; *Miller and Yung*,
2000].
Isotopologues partition differently between phases giving rise to the VPIE. This is most
notable in gas-liquid systems [*Van Hook et al.*, 2001], but also can occur in gas-solid equilibrium.





Both of these may ultimately be important for understanding $\delta^{15}N$ variability in $NO_y$ compounds.
For example, solid-gas VPIE may be relevant for the $HNO_{3(g)} + NH_{3(g)} \longleftrightarrow NH_4NO_{3(s)}$ reaction,
whose temperature-dependent equilibrium can shift dramatically diurnally [*Morino et al., 2006*]
and seasonally [*Paulot et al., 2016*]. It is likely that this VPIE will result in the particle phase
$NO_3^-$ having a different $\delta^{15}N$ value compared to the gas phase $HNO_3$ [Heaton, 1987].
Additionally, possible VPIE occurring during wet and dry deposition, such as $HNO_{3(g)} \rightarrow$
$HNO_{3(aq)}$ may be relevant for $\delta^{15}N$ variations $NO_3^-$ in precipitation *[Freyer et al., 1993].*
Multiphase reactions are not included in RACM since it is only concerned with gas phase
reactions. These effects may be important for accurate $\delta^{15}N$ predictions and should be addressed
in more complex models, but this is a limitation in any "gas phase only" photochemical box
model. Similarly, $NO_y$ aqueous phase reactions, such as $2NO_2 + H_2O \rightarrow HNO_3 + HNO_2$, are not
included in RACM, which may limit $i_N$RACM's ability to accurately predict the $\delta^{15}N$ values of
dissolved $NO_3^-$ in rainfall samples.
## 2.3 Sensitivity analysis: Determining the "reaction relevance" of $NO_y$
isotopologues
The objective of the $i_N$RACM model
is to make predictions about the temporal
and spatial variation of $\delta^{15}N$ value in various
N compounds caused by EIE, KIE, and
PHIFE, and compare them to observations.
Currently, the $\delta^{15}N$ observations are largely
limited to $HNO_3$, as either particulate or
dissolved $NO_3^-$, but there are a few recent
measurements of the $\delta^{15}N$ values of $NO_x$
[*Walters et al., 2018*] and HONO [*Chai and*
*Hastings, 2018*]. The $\delta^{15}N$ values of organic
nitrates and PAN may be made in the not so
distant future, but there is no published data
to date. Thus, a given isotopologue reaction
pair in $i_N$RACM was considered "relevant"
if it significantly changed the $\delta^{15}N$ value
(±1‰) of $NO_x$, HONO, or $HNO_3$. This
relevance was determined by conducting a
sensitivity analysis on the PHIFE, KIE, and
EIE effects for all N reactions. This was

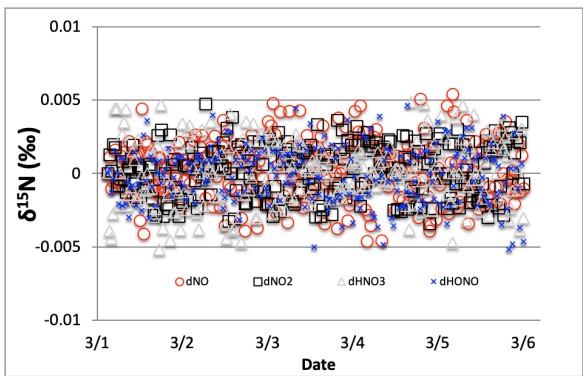

Figure 1. The time evolution of $\delta^{15}N$ values of NO, $NO_2$, HONO, and $HNO_3$, caused by the $NO_3 + NO \rightarrow NO_2 + NO_2$ reaction (R 51, 51$_a$). This reaction only induces a $\delta^{15}N$ variation of +/- 0.005‰ in the relevant compounds. Thus, this reaction is considered irrelevant and $i_N$RACM sets R51a $\alpha = 1.0$.

done by arbitrarily setting $\alpha = 0.98$ ($\varepsilon = -20‰$) for one isotopologue reaction and $\alpha = 1.0$ for all
others, then running a test case. This test case is a 5-day simulation, beginning at 3 AM on
March 1 (2007) and simulates mid-latitude suburban chemistry using the trace gas and
meteorology parameters given in Table S3a-b. This simulation was repeated 96 times until every
N containing reaction was tested. For example, $NO_x$, HONO, or $HNO_3$ $\delta^{15}N$ values are not
sensitive to R51 (Fig. 1). The following section discusses which $i_N$RACM reactions are relevant
and the approaches used to determine the appropriate $\alpha$ values for those reactions. Testing for
isotope mass balance was also performed by hourly summing all N isotopologues, excluding
unreactive $N_2$. Over the course of a five day simulation the total $\delta^{15}N$ value averaged $0.023 \pm$
$0.048‰$, with most of the variance occurring during the initial four hour model spin-up.
Excluding these data points the total $\delta^{15}N$ value averaged $0.018 \pm 0.016‰$, demonstrating
limited impact of rounding errors and effective isotope mass conservation.
2.3.1 PHIFE relevant in the $i_N$RACM
mechanism
10       Only one of the 6 photolysis
reactions involving N compounds was found
to be relevant. $NO_2$ photolysis (R1) had a
significant impact on the $\delta^{15}N$ value of $NO_x$,
HONO, and $HNO_3$ (Fig. 2). The initial
difference between the $\delta^{15}N$ of NO and $NO_2$
values is roughly equal to the arbitrarily set -
20‰ enrichment factor. The nature of the
diurnal oscillation in $\delta^{15}N$ values on the
three relevant $NO_y$ compounds and the
dampening effect over time will be
discussed in the results section.
22       When there is sufficient photolysis
of any single $NO_y$ compound, then the $\delta^{15}N$

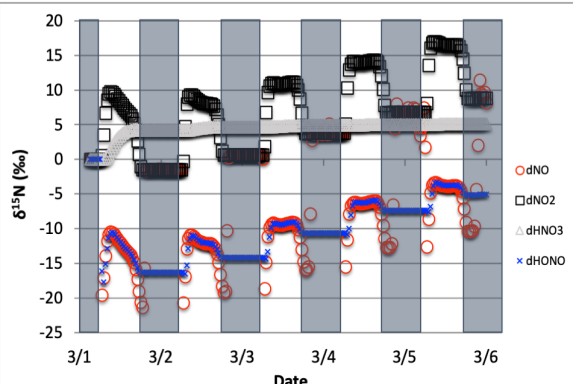

Figure 2. The time evolution of $\delta^{15}N$ values of NO, $NO_2$, $HNO_3$, and HONO caused by PHIFE during $NO_2$ photolysis.

value of that compound tends to significantly change, but often neither the $HNO_3$, HONO, nor
$NO_x$ $\delta^{15}N$ values are affected. For example, the arbitrary $\alpha$ for $NO_3$ photolysis (R7 and R8) alters
the $\delta^{15}N$ value of $HNO_3$ and $NO_x$ by less than 0.1‰ (not shown), but it induces a large diurnal
changes in the $\delta^{15}N$ value of $NO_3$ and $N_2O_5$, with sharp transitions occurring during sunrise and
sunset (Fig. 3). This is easily understood. For our test case, during the day $^{15}NO_3$ would be left
behind because $^{14}NO_3$ is preferentially being photolyzed. The daytime $N_2O_5$ formed from this
$NO_3$ (positive $\delta^{15}N$) and $NO_2$ ($\delta^{15}N \sim 0$) thus has a $\delta^{15}N$ values halfway between these two
reactants (isotope mass balance). However,
there is so little $NO_3$ and $N_2O_5$ during the
day that essentially no $HNO_3$ is being
formed through these precursors and the
$NO_3$ PHIFE is not manifested in the $NO_x$ or
$HNO_3$ $\delta^{15}N$ value. During the night,
photolysis and the PHIFE ceases and any
$NO_3$ and $N_2O_5$ formed by $NO_2$ oxidation
have $\delta^{15}N$ values equal to the $NO_2$.
40       $NO_x$, HONO, and $HNO_3$ are not
sensitive to the other $NO_y$ photolysis
reactions because of this isotope mass
balance effect.
44       $\delta^{15}N_{NOy} = \Sigma f_{NOyi} \cdot \delta^{15}N_{NOyi}$   Eq. (5)
Where $f_{NOyi}$ is the mole fraction of any $NO_{yi}$
compound relative to total $NO_y$, $\delta^{15}N_{NOyi}$ is

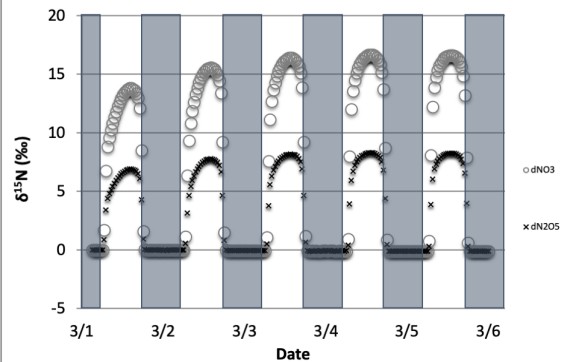

Figure 3. The time evolution of $\delta^{15}N$ values of $NO_3$, and $N_2O_5$ caused by PHIFE during $NO_3$ photolysis.





the $\delta^{15}N$ value of that compound, and $\delta^{15}N_{NOy}$ is the value of total N, which in these simulations is arbitrarily set to 0‰. For an $\varepsilon$ = -20‰ and a threshold of "importance" set to ± 1‰, isotope mass balance requires that $f_{NOyi}$>0.05. Only NO, $NO_2$, HONO, and $HNO_3$ compounds meet this threshold (Fig. 4). All other $f_{NOyi}$ values are an order of magnitude smaller, the largest being $f_{HNO4}$ and it only reaches a maximum value of 0.005. By the end of the second simulation day the $f_{HNO3}$ has approached 1 and effectively minimizes the other $f_{NOyi}$ values because it is the only stable N compound because the other $NO_y$ compounds are very photochemically active. If we exclude this build up in $HNO_3$ from the sum of $NO_y$, then $f_{NO}$, and $f_{NO2}$ (and HONO during some hours, see discussion) become the dominant fractions (Fig. 4) and they control the other $f_{NOyi.}$ Even under this constraint, the $f_{HNO4}$ only reaches 0.001 (Fig. 4). Thus, in $i_N$RACM, the $\alpha$ values of $\alpha_{R4}$- $\alpha_{R8}$ were set equal to 1 and only the $\alpha_{R1}$ was assigned a non-1 value, which was determined using a PHIFE theory (discussed below).

## 2.3.2 KIE relevant in $i_N$RACM mechanism

The KIE for 12 N containing compounds and their 96 reactions were evaluated using the same sensitivity analysis. The vast majority of reactions had little influence on the $\delta^{15}N$ values of $NO_x$, HONO, and $HNO_3$ (Fig. 1). Similar to the photolysis sensitivity, either reaction proximity or isotope mass balance were controlling $\delta^{15}N$ relevance. For example, $NO_2$ + OH is reaction that directly produces a significant fraction of $HNO_3$ and therefore R39 is relevant in the $i_N$RACM mechanism. In contrast, R95 produces very little $HNO_3$ so it has a negligible influence on the predicted $HNO_3$ $\delta^{15}N$ value. Therefore, the only relevant KIE reactions that have $\alpha \neq 1$ in $i_N$RACM mechanism are R39, R91-R97, R48 (Table S2b).

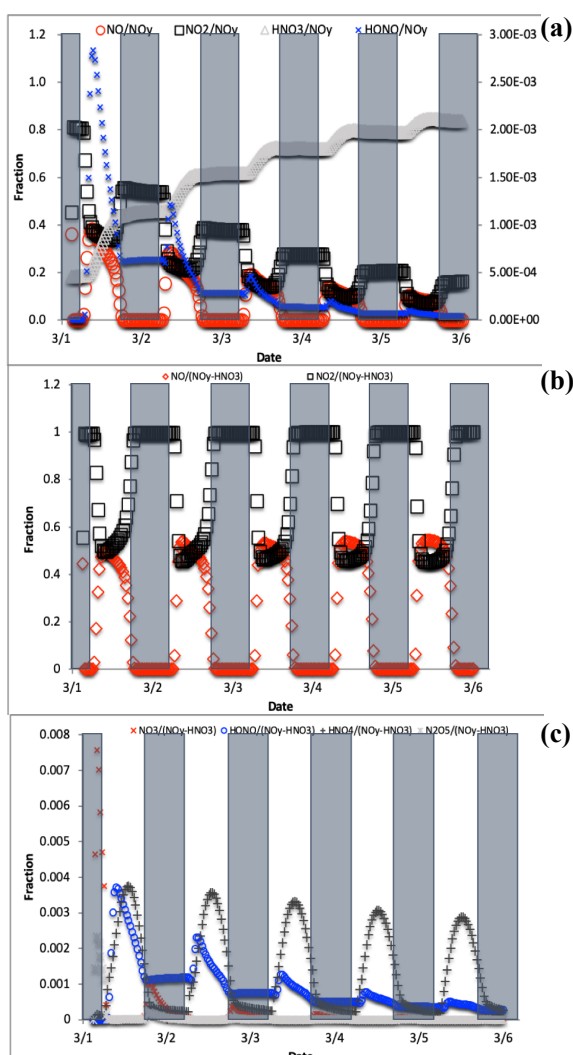

Figure 4. The change in $f_{NO}$, $f_{NO2}$, $f_{NO3}$, and $f_{HONO}$ (right axis) over the 5-day simulation shows the transition from $NO_y$ as mostly $NO_x$ to predominately $HNO_3$ (top, a). For reactive $NO_y$ ($NO_y$ – $HNO_3$) large diurnal changes in $f_{NO}$ and $f_{NO2}$ (middle, b) caused by photolysis minimize the other $f_{NOy}$ values, none of which exceeds 0.01 (bottom, c).

### 2.3.3 EIE relevant in $i_N$RACM mechanism

While some EIE are naturally handled in the $i_N$RACM mechanism, such as the $NO_2$–$NO_3$–$N_2O_5$ equilibrium, other potentially important N isotope exchange reactions are not directly expressed in RACM and must be considered. From a thermodynamic perspective, the EIE for any two N containing compounds can be calculated. The rate at which these compounds can achieve equilibrium, however, needs careful consideration. For example, the EIE for the isotope exchange reaction $NO + {}^{15}HNO_3 \leftrightarrow {}^{15}NO + HNO_3$ has been calculated and measured [*Brown and Begun, 1959*]. Yet, steric considerations would suggest it would be very improbable for a gas phase reaction pathway or transition state to exist where two O atoms and a hydrogen from a $HNO_3$ could quickly migrate to a NO molecule during a collision. The result is that isotope exchange for this gas phase reaction is likely kinetically too slow to be relevant but is valid in a highly concentrated liquid phase [*Brown and Begun, 1959*]. The larger the N containing molecule the more difficult it is to envision gas phase EIE occurring on a timescale comparable to the residence time tropospheric N of about a week. On the other hand, the isotope exchange reaction $NO + {}^{15}NO_2 \leftrightarrow {}^{15}NO + NO_2$ rapidly occurs [*Sharma et al.*, 1970] because it can form an ONONO ($N_2O_3$) stable intermediate. As such, $i_N$RACM only considers N isotope equilibrium between NO, $NO_2$, $NO_3$, and $N_2O_5$. Since the latter 3 compounds are already *chemically* equilibrated in RACM, they are by default isotopically equilibrated in $i_N$RACM. Therefore, the only new isotope exchange reaction added to $i_N$RACM was $NO + {}^{15}NO_2 \leftrightarrow {}^{15}NO + NO_2$ (R238, R238a).

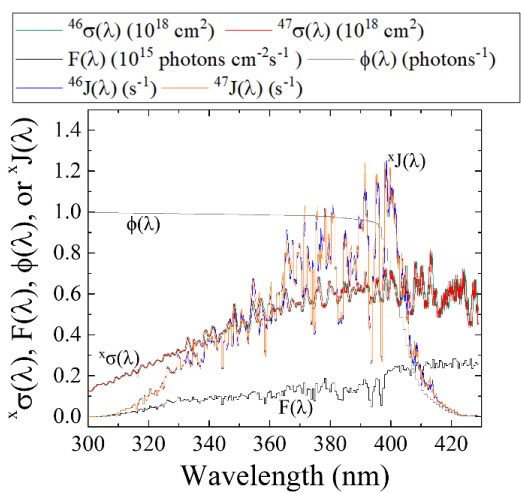

Figure 5. Literature reported ${}^{46}\sigma(\lambda)$ [*Vandaele et al., 2002*] $F(\lambda)$ (at SZA of 60°; TUV model), and $NO_2$ $\phi(\lambda)$ [*Roehl et al., 1994*] and calculated ${}^{47}\sigma(\lambda)$ derived from the ZPE shift model for wavelengths relevant for tropospheric conditions for $NO_2$ photolysis. From these parameters, both ${}^{46}J(\lambda)$ and ${}^{47}J(\lambda)$ have been calculated (Eq. (2)).

### 2.4 Isotopologue fraction factors ($\alpha$) used in $i_N$RACM

In this section we discuss the methodology used to determine the values for the relevant PHIFE, KIE, and EIE. These are reactions R1, R39, R48, R91-R97, and R238.

#### 2.4.1 PHIFE derived $\alpha$ used in the $i_N$RACM mechanism

The PHIFE for R1 was calculated using existing $NO_2$ experimental photolysis cross-section of ${}^{14}NO_2$ for tropospheric relevant wavelengths (300 to 450 nm) [*Vandaele et* al., 2002]. Using the experimentally determined $\Delta$ZPE for the ${}^{15}NO_2$ isotopologue of 29.79 cm${}^{-1}$ [*Michalski et al., 2004*], the ${}^{47}\sigma(\lambda)$ was blue shifted by roughly 0.3 nm from the experimentally measured



$^{46}\sigma(\lambda)$ [*Vandaele et al., 2002*] (Fig. 5).    The wavelength dependent actinic flux, F($\lambda$), was taken
from the TUV model (NCAR) for solar zenith angles from 0 to 90° in 15° increments. The $\phi(\lambda)$
values were taken from experimental data at 298 K [*Roehl et al., 1994*], and it was assumed that
there is no significant quantum yield isotope effect. Based on these assumptions the $^{46}J(\lambda)$ and
$^{47}J(\lambda)$ values were calculated (Fig. 5). An important feature of $NO_2$ the wavelength dependent J
include a peak near 390-400 nm that subsequently decreases at longer wavelengths until $NO_2$
photolysis ceases beyond 420 nm due to a $\phi = 0$ beyond this wavelength [*Roehl et al., 1994*].
Overall, the $NO_2$ PHIFE $\alpha$ value was found to be consistent for the wide range of solar zenith
angles, ranging between 1.002 to 1.0042 with higher values occurring at lower solar zenith
angles. We used an $\alpha = 1.0042$ for daylight hours.
2.4.2 KIE derived $\alpha$ used in the $i_N$RACM mechanism
2.4.2.1 KIE for the NO + $O_3$ reaction
15        The $^{15}\alpha_{48}$ for the reaction NO + $O_3 \rightarrow NO_2 + O_2$ reaction was determined by *ab initio*
calculations [*Walters and Michalski, 2016*].  Generally, in a normal KIE the heavy $^{15}$NO would
react with $O_3$ slower than the light $^{14}$NO, which consistent with the calculated effect, however, it
is relatively small ($\varepsilon$ = -6.7‰ at 298 K).  The $^{15}\alpha_{48}$ was determined to have the following
temperature dependent relationship [*Walters and Michalski*, 2016] over the temperature range of
220 to 320 K (Eq. (6)):
$\alpha_{48} = (0.9822*\exp(3.3523/T)$          Eq. (6)
2.4.2.2 KIE for the $NO_3$ + VOC reactions
25        The most influential reactions that impacted the $\delta^{15}$N of $HNO_3$ were the three reaction
pathways that generate $HNO_3$.  This is because the isotope effect associated with this last step is
largely retained in the product $HNO_3$ because photolysis of $HNO_3$ back into photochemically
active compounds that could re-scramble N isotopes is slow, effectively "locking in" these final
isotope effects.  Two gas phase reactions groups are important for $HNO_3$ production.  Nitric acid
is produced mainly by R39 during the daytime [*Seinfeld and Pandis, 1998*] but this reaction is
treated as an EIE as discussed below in the EIE section. During the nighttime, when the
photolysis sink for $NO_3$ vanishes, $NO_3$ can react with VOCs to form $HNO_3$ via hydrogen
abstraction reactions [*Atkinson, 2000*]. Any individual $NO_3$ + VOC reaction had a small
"relevance" for the $\delta^{15}$N values of $NO_x$, and $HNO_3$, but given there are 7 such reactions (R91-
R97) their sum may be important.
36        The KIE for each of the $NO_3$ +VOC$\rightarrow$ $HNO_3$ reaction (R91-R97) was determined by
assuming collisional frequency was the key KIE factor in such reactions.  In these reactions
(R91-R97) $NO_3$ abstracts a hydrogen from a hydrocarbon, acting though a transition state
involving the oxygen atoms in the nitrate radical C--H--$ONO_2$.  Since N is not directly
participating in the bond formation it is classified as a secondary KIE [*Wolfsberg*, 1960].
Secondary KIE are typically much smaller than primary KIEs that occur at bond
breaking/forming positions within a molecule [*Wolfsberg*, 1960].  Therefore, we assumed that
the secondary KIE was negligible and did not factor into the $\alpha$ values for these 7 reactions. On
the other hand, isotope substitution does change the relative rate of collisions for N
isotopologues because of the change in molecular mass.  The collisional frequency (Eq.7) for any
of the $NO_3$ + VOC reaction pair was calculated assuming a hard sphere approximation via





$$A = \left[\frac{8kT}{\pi\mu}\right]^{1/2} \pi d^2 \qquad \text{Eq. (7)}$$

here $\mu$ is the reduced mass of either $NO_3$ or $^{15}NO_3$ and the specific hydrocarbon in a given reaction (R91-R97). When taking the isotopologue collision ratio, the constants, collision cross-section ($d^2$), and temperature cancel out giving a temperature independent KIE of

$$\alpha = \frac{k_{15}}{k_{14}} = \frac{A_{15}}{A_{14}} = \sqrt{\frac{\mu_{15}}{\mu_{14}}} \qquad \text{Eq. (8)}$$

The $\alpha$ for each $NO_3$ + VOC reaction (R91-R97) as calculated using the hydrocarbon mass (Table S1b) and the $NO_3$ isotopologue masses (62, 63 amu) and using Eq. (8).

### 2.4.3 EIE derived $\alpha$ used in the $i_N$RACM mechanism

#### 2.4.3.1 EIE of NO + $NO_2$ exchange

The NO + $NO_2$ exchange was added to $i_N$RACM by defining a forward and reverse reaction (R238, R238a) and an equilibrium constant $K_{238} = k_{238}/k_{238a} = \alpha$. The forward rate constant ($k_{238}$) was based on the NO-$NO_2$ isotope exchange rate determined by Sharma et al. ($3.6*10^{14}$ cm$^3$ s$^{-1}$ molecule$^{-1}$). The reverse rate was calculated using $k_{238} = k_{238a}/\alpha_{238}$. The temperature-dependent for EIE of NO + $NO_2$ exchange (Eq. 9) was calculated using quantum mechanical techniques [*Walters and Michalski*, 2015] that matched well with recent experimental values [*Walters et al.*, 2016].

$$\alpha_{238} = 0.9771*\exp(18.467/T) \qquad \text{Eq. (9)}$$

#### 2.4.3.2 EIE used in the $NO_2$ + OH reaction

The $^{15}\alpha_{39}$ for the $NO_2$+OH+M $\rightarrow$ $HNO_3$ reaction (R39) was determined by assuming equilibrium between $NO_2$ and $HNO_3$. The third body and the negative temperature dependence of the rate constant shows that, similar to $O_3$ formation, this reaction is an association reaction [*Golden and Smith, 2000*]. It proceeds through an excited intermediate, *$HNO_3$, that can undergo collisional deactivation by a third body M (Eq.10).

$$NO_2 + OH \overset{k_f}{\underset{k_r}{\rightleftarrows}} *HNO_3 \overset{k_d}{\rightarrow} HNO_3 \qquad \text{Eq. (10)}$$

in which $k_f$ and $k_r$ are the forward and reverse rate constants for the association step and $k_d$ is the rate constant for collisional deactivation. The $HNO_3$ production rate constant is then $k_f k_d[M]/k_r = K_{eq}k_d[M]$. This general form can be used to write two isotopologue equilibrium constants K

$$K_{39} = [*HNO_3]/([NO_2][OH]) = k_{39f}/k_{39r} \qquad \text{Eq. (11)}$$
$$K_{39a} = [*H^{15}NO_3]/([^{15}NO_2][OH]) = k_{39af}/k_{39ar} \qquad \text{Eq. (12)}$$

Since •OH is not participating in the N isotope chemistry, these two EIE effectively reduces the isotope chemistry to the temperature dependent $^{15}N$ EIE





$^{15}NO_2 + {}^*HNO_3 \longleftrightarrow NO_2 + {}^*H^{15}NO_3$      Eq. (13)

$K_{39a}/K_{39} = \alpha_{HNO3/NO2} = \beta_{HNO3}/\beta_{NO2}$      Eq. (14)

The fundamental vibration frequencies for $HNO_3{}^*$ were taken to be the same as ground state $HNO_3$, similar to RRKM theory approaches used to calculate the uni-molecular decay rate of $HNO_3{}^*$ [*Golden and Smith, 2000*]. The temperature-dependent $\beta_{HNO3}$ and $\beta_{NO2}$ values for this exchange were taken from [*Walters and Michalski, 2015*]. Since the reaction has a negative activation energy and has a fairly rapid rate constant at 101 kPa, ($1 \times 10^{11}$ $cm^{-3}$ $s^{-1}$) and the isotope effect due to the collisional deactivation frequency (Eq. 7) is minimal (~2‰) compared to the equilibrium effect (~40‰), the deactivation rate constants $k_d$ were set equal ($k_{d14}/k_{d15}=1$). Setting $k_{r14}= k_{r15}$, and using the $\alpha_{HNO3/NO2}$ equilibrium value the $k_{39a}$ for the $^{15}NO_2 + OH \rightarrow H^{15}NO_3$ reaction is

$K_{39a} = \alpha_{HNO3/NO2} (K_{39})$      Eq. (15)

The temperature dependence of $\alpha_{HNO3/NO2}$ is derived from the tables in [*Walters and Michalski, 2015*] and $\alpha_{39}$ is then:

$\alpha_{39} = (0.973 {}^* exp(19.743/T)$      Eq. (16)

For typical tropospheric temperatures the $\alpha_{HNO3/NO2}$ 1.040 suggesting the $\delta^{15}N$ of $HNO_3$ produced by the $NO_2 + OH$ reaction will be +40‰ relative to tropospheric $NO_2$. This $\alpha$ value is larger and opposite the sign of the $^{15}\alpha = 0.9971$ assumed by Freyer et al. (*1993*). Freyer's $\alpha$ was approximated by the using reduced mass of the $OH-NO_2$ activated complex. There two problems with this approach. First, the activation complex's reduced mass approximation should be viewed in terms as the *decomposition* rate constant, not the product formation rate constant as assumed by Freyer, because transition state theory assumes equilibrium between the stable *reactants* and the transition state [*Bigeleisen and Wolfsberg*, 1958; *Wolfsberg et al.*, 2010]. In other words, Freyer's $\alpha = 0.9971$ should indicate that the $^{15}NO_2-OH$ decomposes more slowly than $^{14}NO_2-OH$ and therefore more likely to form $HNO_3$ at +2.9‰ (not -2.9‰ determined in Freyer). Secondly, the reduced mass approximation of the complex pair ignores the thermodynamic contribution of the reactants and the vibrations in the transitions state other than the bond forming (imaginary) vibration. Our approach overcomes both of these assumptions and incorporates the temperature dependence of the EIE for this reaction.

2.4.3.3 EIE used in heterogeneous reactions of $N_2O_5$

During the nighttime, the heterogeneous $HNO_3$ formation pathway becomes important [*Chang et al., 2011; Dentener and Crutzen, 1993; Riemer et al., 2003*]. During the night, NO is nearly completely oxidized to $NO_2$ leading to the build-up of the $NO_3$ radical (R48), the formation of $N_2O_5$ (R53), and heterogeneous $N_2O_5$ hydrolysis becomes a major source of $HNO_3$ production (discussed below). This is particularly true in regions that have high $NO_x$ mixing ratios and large aerosol surface areas such as urban centers [*Chang et al., 2011; Riemer et al., 2003*]. In order to assess the $^{15}N$ partitioning of this reaction pathway, both EIE and KIE were considered.

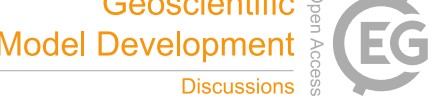

It was assumed that the fractionation factor for the $N_2O_5 \rightarrow 2HNO_3$ reaction was mainly controlled by nighttime equilibrium between $N_2O_5$ and $NO_2/NO_3$ (R53, R54). When factoring the isotopologue dynamics, this equilibrium can be viewed as an EIE via

$$^{15}NO_2 + NO_3 \longleftrightarrow O_2^{15}N\text{--}O\text{--}NO_2 \longleftrightarrow NO_2 + {}^{15}NO_3 \qquad \text{Eq. (17)}$$

here $^{15}N_2O_5$ is represented as the transition state $O_2^{15}N\text{--}O\text{--}NO_2$ to highlight the relative ease of N isotope exchange via oxygen migration during $N_2O_5$ formation and decomposition. The symmetry of $^{15}NNO_5$ and $N^{15}NO_5$ is also why they were not treated as isotopomers since they are structurally identical.

The $N_2O_5$ equilibrium in the RACM model is dealt with as a forward reaction R53 ($k_{53}$) and a decomposition reaction R54 ($k_{54}$) that are derived from the measured equilibrium constant ($K_{53}$) = ($k_{53}/k_{54}$). In $i_N$RACM the $N_2O_5$ isotopologue has 2 formation pathways, with two forward rate constants ($k_{53\ a,b}$) and two decomposition rate constants ($k_{54\ a,b}$) that were used to write their respective equilibrium constants K

$$^{15}NO_2 + NO_3 \longleftrightarrow {}^{15}NNO_5 \quad (K_{53a} = k_{53a}/k_{54a}) \quad \text{Eq. (18)}$$
$$NO_2 + {}^{15}NO_3 \longleftrightarrow {}^{15}NNO_5 \quad (K_{53b} = k_{53b}/k_{54b}) \quad \text{Eq. (19)}$$

Dividing $K_{53a}$ and $K_{53b}$ by $K_{53}$ yields isotopologue product and reactant ratios that can be evaluated using $\beta(\alpha)$ values from Walters and Michalski (*2015*). These were used to determine the $\alpha$ value for the $N_2O_5$ isotopologue equilibrium, which are simply a function of the formation and decomposition rate constants and temperature

$$K_{53a}/K_{53} = (^{15}NNO_5/N_2O_5)(NO_2/^{15}NO_2)(NO_3/NO_3) = \beta_{N2O5}/\beta_{NO2}$$
$$= \alpha_{N2O5/NO2} = k_{53a}/k_{53} \times k_{54}/k_{54a} \qquad \text{Eq. (20)}$$
$$K_{53b}/K_{53} = (^{15}NNO_5/N_2O_5)(NO_3/^{15}NO_3)(NO_2/NO_2) = \beta_{N2O5}/\beta_{NO3}$$
$$= \alpha_{N2O5/NO3} = k_{53b}/k_{53} \times k_{54}/k_{54b} \qquad \text{Eq. (21)}$$

The $N_2O_5$ decomposition rate constants were arbitrarily set to be equal ($k_{54} = k_{54a} = k_{54b}$) and the decomposition rate constants were then derived using the temperature dependent $\alpha$ values

$$k_{53a} = k_{53}(\alpha_{N2O5/NO2}) \quad \alpha_{N2O5/NO2} = 1.0266 \text{ (298 K)} \qquad \text{Eq. (22)}$$
$$k_{53b} = k_{53}(\alpha_{N2O5/NO3}) \quad \alpha_{N2O5/NO3} = 1.0309 \text{ (298 K)} \qquad \text{Eq. (23)}$$

The $\alpha$ for doubly substituted $^{15}N_2O_5$ isotopologue was determined using $\alpha = \beta_{15N2O5}/\beta_{NO2}\beta_{NO3}$ and the value for $\beta_{15N2O5}$ (1.272) was approximated using the principle of the geometric mean [*Bigeleisen*, 1958; *Snyder et al.*, 1999], yielding a temperature independent $\alpha = 1.057$. However, the $N_2O_5$ system is insensitive to this value because the low probability of a $^{15}N +{}^{15}N$ reaction ($1.5 \times 10^{-5}$) relative to a $^{14}N + {}^{15}N$ reaction ($4 \times 10^{-3}$), thus the small temperature dependence was also ignored.

Because RACM is a gas phase chemical mechanism, it does not include heterogeneous reactions of $N_2O_5$ on aerosols, which would limit $i_N$RACM to accurately predict the $\delta^{15}N$ values, particularly at night. Gas chemical mechanisms are often used in larger 1, 2, and 3-D chemical



transport models that usually also include aerosol modules that calculate heterogeneous
chemistry using inputs from the gas phase chemical mechanism (i.e. $N_2O_5$ concentrations).
However, if the objective is to use a 0-D chemical box model to simulate local chemistry the
$N_2O_5$ heterogeneous hydrolysis will need to be included. $i_N$RACM was modified to use a first
order rate constant to calculate $N_2O_5$ heterogeneous hydrolysis [*Yvon et al. 1996; Riemer et al.,*
*2003*]. The rate constant is a function of $N_2O_5$ molecular speed (c), the $N_2O_5$ uptake coefficient
($\gamma$) and the aerosol surface area density S.
$-dN_2O_5/dt = d0.5HNO_3/dt = k_{N2O5}[N_2O_5] = R239$ $\qquad$ $k_{N2O5} = \frac{1}{4}c\,\gamma\,S$ Eq. (24)
$\qquad$ The $k_{N2O5}$ values were assessed based
on the different pollutant loadings and
emission scenarios (Fig. 6). The $k_{N2O5}$ was
calculated as a function of $\gamma$ [*Anttila et al,*
*2006; Bertram & Thornton. 2009; Davis et al.,*
*2008; Riemer et al., 2003; Riemer et al., 2009*]
and S [*Cai et al., 2018; Kuang et al., 2010;*
*McMurry et al., 2005; Petäjä et al., 2009; Qi*
*et al., 2015*] values that span clean to highly
polluted environments. This range yielded
$k_{N2O5}$ = 1, 0.1, and 0.01 for high, medium, and
low polluted environments (Fig. 6).

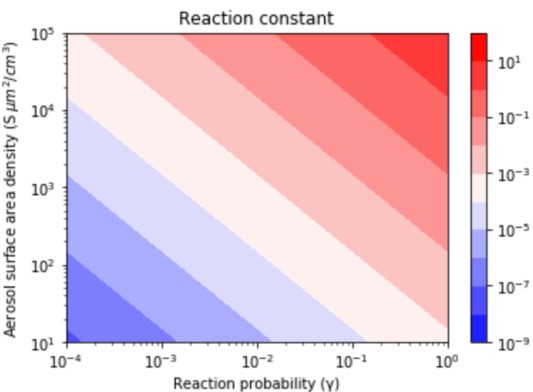

$\qquad$ Only the uptake coefficient ($\gamma$) and
molecular speed (c) could have a KIE during
aerosol uptake of $N_2O_5$ (R239, R239a, R239b).
The $\gamma$ term was ignored because *ab initio* work
suggests that $N_2O_5$ hydrolysis activates
through hydrogen bonding between water
molecules on the aerosol surface and O atom
in the $N_2O_5$ [*Snyder et al.*, 1999] making it a
secondary (small) KIE for N. The c term is a
function of the root of the $N_2O_5$ molecular
mass and when the ratio is taken there is no
temperature dependence yielding $\alpha_{239a}$ =
$(108/109)^{0.5} = 0.995$ and $\alpha_{239b} = (108/110)^{0.5} = 0.9909$.

Figure 6. Contour lines of the same $k_{N2O5}$ values as a function of $\gamma$ and S values. The $\gamma$ values depend on aerosol composition and range from $3.8 \times 10^{-5}$ (relatively dry sulfuric acid) to 1 (aqueous aerosol in the winter polar stratosphere). S values are a function of aerosol number density and size distribution and range from 52 (low scavenging rate, low particle growth rate) to 1140.1 (high scavenging rate, high particle growth rate).

An online interactive version of this $i_N$RACM model is available for public use at
https://mygeohub.org/tools/sbox/

2.4.4 $i_N$RACM simulations
$\qquad$ A number of $i_N$RACM simulations were run with two different purposes. The first set of
simulations iteratively changed the $\alpha$ values from 1 to their values discussed above. These
simulations aimed at investigating the importance of each $\alpha$ as they aggregated together. These
include photolysis only, Leighton cycle, daytime chemistry, night-time chemistry, and full
chemistry using the same test case (Table S3a-f). The second set of simulations replicated the



test case simulations (Table S4a-b, 5a-b) detailed in Stockwell (1997) and other pollution scenarios (Table S8). These were run with all $\alpha$'s activated but with varied initialized chemistry and primary pollutant emissions.

## 3.0 Results and Discussion

It is important to first test $i_N$RACM by turning on and off individual relevant isotope effects and then combining their cumulative effects. This is advantageous relative to simply running the full mechanism under different pollution scenarios because it would be a challenge to disentangle which isotope effects in the full mechanism were mainly responsible for $\delta^{15}N$ change in $NO_x$, HONO, or $HNO_3$ without such a systematic investigation. For example, it is likely that the $\delta^{15}N$ value of $NO_2$ will be a significant factor in the $\delta^{15}N$ value of $HNO_3$ because it is the reactant in R39 and R239. Thus, understanding which isotope effects control the $\delta^{15}N$ of $NO_2$ helps with interpreting the $\delta^{15}N$ value of $HNO_3$ and vice versa. Thus, this discussion section is divided into 3 sections. The first is the examination of the relevant isotope effects occurring during daytime photochemistry and their impact on $NO_x$, HONO, and $HNO_3$ $\delta^{15}N$ values. Secondly, is the examination of the relevant isotope effects occurring during nighttime chemistry (EIE and KIE) and their effect on $NO_x$, HONO, and $HNO_3$ $\delta^{15}N$ values. These first two discussion sections focus mainly on the relative importance of each isotope effect when the photochemical conditions are constant. Finally, the full $i_N$RACM mechanism will be tested under different atmospheric conditions such as variations in trace gas concentrations, aerosol loading, and hours of sunlight. This tests how changes in photochemical oxidation pathways results in difference in the $\delta^{15}N$ values of $NO_x$, HONO, and $HNO_3$.

## 3.1 The $\delta^{15}N$ of $NO_x$, HONO, and $HNO_3$ due to daytime chemistry

The role that daytime chemistry plays in determining the $\delta^{15}N$ values of $NO_x$, HONO, and $HNO_3$ was investigated by iteratively adding relevant fractionation factors to $i_N$RACM. The sensitivity of $NO_x$, HONO, and $HNO_3$ $\delta^{15}N$ values to $NO_2$ photolysis (R1a) was tested. The initial trace gas concentrations and emissions were set to the March 1 test cases (Table S3 a-f) and simulations were run with, and without, NO emissions. All subsequent test simulations will also use the March 1 test case in order to have a consistent comparison of $\delta^{15}N$ values between different simulations. It is noted that the initial $HNO_3$ and $O_3$ mixing ratios are set to zero and that the start time of the simulations is 3 a.m. The main daytime only effects will be $NO_2$ photolysis (R1), $O_3$ oxidation (R8) and reaction OH (R39) since both photolysis and OH chemistry is only relevant during the daytime. However, $NO_x$ isotope exchange and NO + $O_3$ will also play a vital role despite no being exclusively daytime reactions.

3.1.1 The $\delta^{15}N$ values of $NO_x$, HONO, and $HNO_3$ due to the photolysis only



The simulations with only R1 isotope effect activated (with $NO_x$ emissions) shows a clear diurnal cycle in $NO_x$ and HONO $\delta^{15}N$ values and a weekly trend moving towards an approximate steady-state for $HNO_3$ $\delta^{15}N$ values, which can be explained by the PHIFE (Fig. 7a). Initially all $NO_y$ has $\delta^{15}N$ of zero (by default) and there is no photolysis at 3 am. At sunrise the $\delta^{15}N$ value of $NO_2$ goes negative and NO value positive since $^{15}NO_2$ is preferentially photolyzed ($\alpha_{R1}$ = 1.0042). The difference between the $\delta^{15}N$ values of NO and $NO_2$ ($\Delta\delta^{15}N_{NO-NO2}$ = $\delta^{15}N$ NO - $\delta^{15}N$ $NO_2$) at all times during the day is 4‰, which is the $\varepsilon_{R1a}$ value. During the night both the NO and $NO_2$ $\delta^{15}N$ values approach 0‰ because most NO is oxidized to $NO_2$ and NO emissions (0‰) dominate the NO nighttime budget (relative to residual day NO). Over the weeklong simulation, the $NO_x$ $\delta^{15}N$ value slowly increases by about one per mil. This is because $^{15}N$ depleted $NO_2$ is converted into $HNO_3$ leaving the residual $NO_x$ $^{15}N$ enriched. This is also the reason for the $\delta^{15}N$ values of $HNO_3$ that initially mimic the

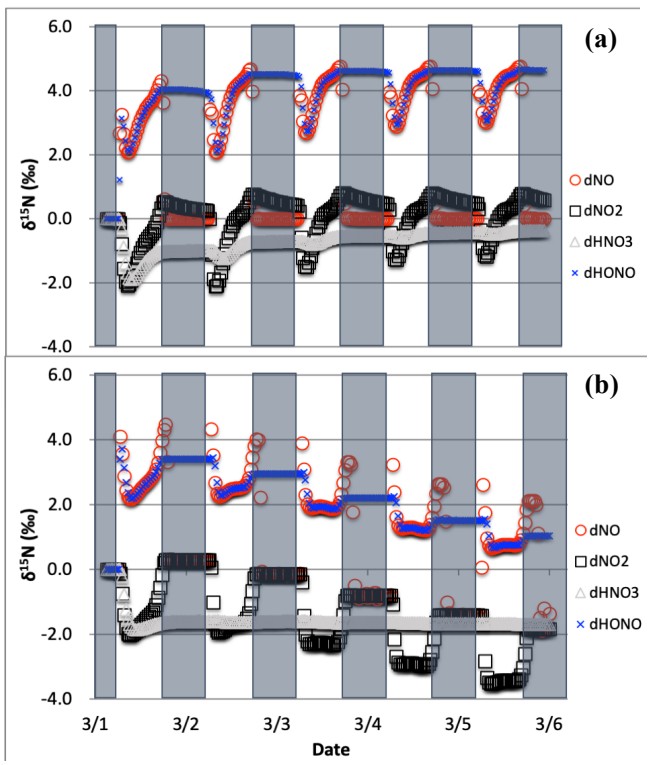

Figure 7. The $\delta^{15}N$ values of NO (O), $NO_2$, (□) HONO (x), and $HNO_3$ (△) with only the photolysis isotope fractionations active. The 5-day simulation was under the conditions list in Table S3a-b. The gray boxes span night hours and the white span daytime. The top (a) is the simulation with $NO_x$ emissions and the bottom (b) is without $NO_x$ emissions.

daytime $NO_2$ values and trends towards 0‰ by the end of the simulation week. The $\delta^{15}N$ values of HONO mimics the NO values during the daytime since the main reaction pathway forming HONO is OH + NO, which peaks in the morning (~10:00). HONO retains the evening $\delta^{15}N$ values through the night since most of the HONO is destroyed in the afternoon via photolysis and again follows NO $\delta^{15}N$ the next morning as its production again reaches a maximum (Fig. 7).

The simulation without NO emissions shows a similar behavior but with some clear differences relative to the emission case. The $NO_x$ and HONO $\delta^{15}N$ values exhibit the same diurnal $\Delta\delta^{15}N_{NO-NO2}$ = 4‰ value. Unlike the emission case, however, the diurnal $NO_x$ $\delta^{15}N$ value peaks and troughs trend downward during the week-long simulation, with NO approaching 0‰ and $NO_2$ approaching -4‰. The $HNO_3$ $\delta^{15}N$ values reach roughly a steady state value of -1.7‰ after about a day and $NO_x$ is ~ -1.8‰ (Fig. 7b). This difference between the emission and non-emission case is a consequence of isotope mass balance ($f_x$ = mole fraction of compound x relative to total $NO_y$).

$\delta^{15}N_{total} = 0 = f_{NOx} \bullet \delta^{15}N_{NOx} + f_{HNO3} \bullet \delta^{15}N_{NHNO3} + f_{ONIT} \bullet \delta^{15}N_{ONIT}$        Eq. (25)
The positive $\delta^{15}N$ $NO_y$ compound that effectively offsets the -1.7‰ in $HNO_3$ and -1.8‰ in $NO_x$
is organic nitrate that is +2‰ and makes about half the $NO_y$ pool and is roughly equal to $HNO_3$ +
$NO_x$ ($f_{NOx}$ = 0.11, $f_{HNO3}$ = 0.36, $f_{ONIT}$ = 0.53). In the $NO_x$ emission case only about 5% of $NO_y$ is
as organic nitrate ($f_{NOx}$ = 0.17, $f_{HNO3}$ = 0.78, $f_{ONIT}$ = 0.05) indicating a shift in oxidation pathways
when NO and VOCs are emitted during
the simulation relative to when they are
not. In the emissions case the $NO_x$ mixing
ratios at the end of the simulation are
actually slightly higher than their initial
ratios, in contrast to the no $NO_x$ emission
case where 90% of $NO_x$ has been lost via
oxidization into organic nitrate and $HNO_3$.
This loss of N in the no emission scenario
effectively shuts down the oxidation
chemistry. For example, the day 5 mixing
ratio of $O_3$ is 45 ppb$_v$ (reasonable) for the
emission case but only 2 ppb$_v$ for the non-
emission case (unreasonable). Therefore,
we exclude no-emission simulations for
the chemistry analysis discussed in this
section and restrict them to the no
emission simulations to 48 hours in the
final test case analysis (See section 4).

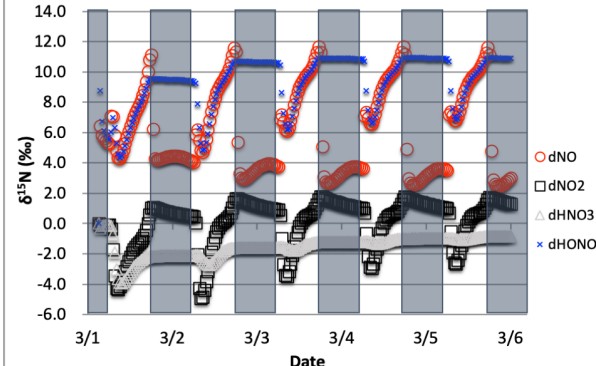

Figure 8. The $\delta^{15}N$ values of $NO_x$, HONO, and $HNO_3$
when isotope effects associated R1 and R48 are
combined, with $NO_x$ emission. The 5-day simulation
was under the conditions list in Table S3a-b. The
diurnal patterns are reflecting the relative importance of
photolysis and $O_3$ chemistry during the day and night.

3.1.2 The $\delta^{15}N$ values of $NO_x$, HONO, and $HNO_3$ due to the combined Leighton cycle
The simulations with both $NO_2$ photolysis (R1) and $O_3$ + NO (R48) isotope effects active
shows similar diurnal and weekly trends as the photolysis only simulations, they are just slightly
amplified (Fig. 8). The daytime $\Delta\delta^{15}N_{NO-NO2}$ is now ~ 9.5‰, which is close to the additive of the
two isotope effects ($\epsilon_{48a}$ = -6.7‰, $\epsilon_{R1a}$ = 4.2‰). This is logical since $^{15}NO$ is reacting with $O_3$
slower than $^{14}NO$, preferentially leaving behind $^{15}NO$ and thus the higher NO $\delta^{15}N$ value. The
$HNO_3$ $\delta^{15}N$ values reach the mean of the daytime $NO_2$ $\delta^{15}N$ values via the $NO_2$ + OH reaction.
The slight (1‰) upward trend of $NO_x$ and $HNO_3$ are due to isotope mass balance as detailed in
the photolysis only case. Similar to the photolysis only case the $\delta^{15}N$ of HONO is mimicking
daytime NO $\delta^{15}N$ values.
3.1.3 The $\delta^{15}N$ values of $NO_x$, HONO, and $HNO_3$ due to the combined Leighton cycle and $NO_x$
isotope exchange
The $\delta^{15}N$ values of $NO_x$ produced when both the Leighton cycle and $NO_x$ isotope
exchange are active exhibit a very dynamic diurnal range that is a function of the $NO_x$ mixing
ratios. At high $NO_x$ mixing ratios (150 ppb, 1/3 NO, 2/3 $NO_2$, Fig. 9a) the $\Delta\delta^{15}N_{NO-NO2}$ is -40‰
at night as expected for $NO_x$ isotopic equilibrium ($\epsilon_{NO/NO2}$ = -40‰ at 298K). During the daytime
the $\Delta\delta^{15}N_{NOx}$ shifts -30 to -35‰ as the photolysis and $O_3$ isotope effects begin to influence the



$\Delta\delta^{15}N_{NO-NO2}$. HNO$_3$ $\delta^{15}$N values during the high NO$_x$ mixing ratio simulation initially follow the
$\delta^{15}$N of NO$_2$ (via NO$_2$ + OH) before approaching 0‰, the defined NO$_x$ source values.
At low NO$_x$ mixing ratios (1.5 ppb,
1/3 NO, 2/3 NO$_2$, Fig. 9c) the $\Delta\delta^{15}N_{NO-NO2}$
and HNO$_3$ $\delta^{15}$N is very different from the
high NO$_x$ simulation. The nighttime
$\Delta\delta^{15}N_{NO-NO2}$ ranges from -15 to -20‰ and
during the daytime it is around +7‰, while
the HNO$_3$ $\delta^{15}$N values hover around zero
throughout the simulation. The difference
between the NO$_y$ $\delta^{15}$N values in the high and
low NO$_x$ cases can be explained as a
competition between the NO$_x$ EIE and the
Leighton isotope effect. At high NO$_x$ mixing
ratios, the NO$_x$ EIE achieves equilibrium
quickly at night ($\Delta\delta^{15}N_{NO-NO2}$ = -40) because
the rate of NO$_x$ isotope exchange (R238) is
proportional to its concentration. In contrast,
isotope exchange is slow in the low NO$_x$
case and the time scale to reach equilibrium
is much longer.   Indeed, at the low NO$_x$
mixing ratios the nighttime equilibrium only
reaches about 40-50% of completion by 6:30.
Afterwards sunlight begins to erase the NO$_x$
EIE effect until around noon when the $\delta^{15}$N
values of NO is mostly due to the Leighton
effect and only a small contribution from
EIE (about 5%).   For intermediate NO$_x$
mixing ratio case (15 ppb, 1/3 NO, 2/3 NO$_2$,
Fig. 9b) the diurnal and week-long NO$_y$ $\delta^{15}$N
trends fall somewhere in between the high
and low NO$_x$ simulations.
The changes in $\delta^{15}$N values of HNO$_3$
during the March 1 simulations at differing
NO$_x$ mixing ratios can be explained in terms
of HNO$_3$ production pathways. Over the
course of day 1 the $\delta^{15}$N of HNO$_3$ mirrors
that of NO$_2$ because HNO$_3$ produced by NO$_2$
+ OH (R39), thus the product HNO$_3$ $\delta^{15}$N
values are similar to those in NO$_2$.   This
varies depending on the NO$_x$ mixing ratio
scenario for two reasons. First, as the NO$_x$
mixing ratio gets bigger, the closer the NO$_x$
gets to achieving the EIE and the bigger the

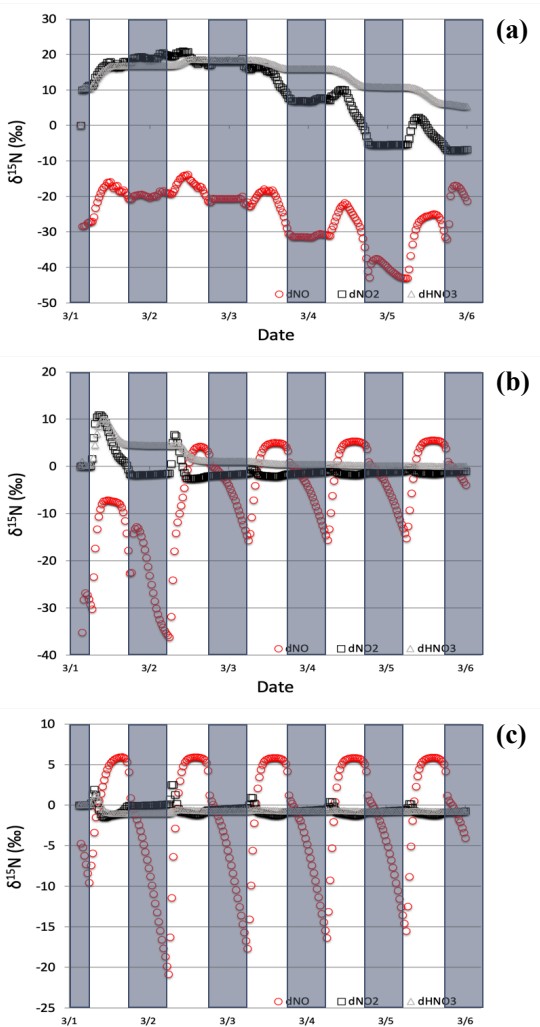

Figure 9. The $\delta^{15}$N values of NO$_x$ and HNO$_3$ when isotope effects in R1, R48, and R238 are included under high (top, a), medium (middle, b), and low (bottom, c) NO$_x$ scenarios. The 5-day simulation was under the conditions list in Table S3d-f. The NO$_y$ $\delta^{15}$N values are mainly controlled by NO$_x$ isotope exchange (R238) under high NO$_x$ conditions and Leighton (R1 + R58) under low NO$_x$ conditions.

split between NO and NO$_2$ $\delta^{15}$N values (40‰ versus 10‰ for Leighton+O$_3$). Secondly,





differences in the amount of $NO_x$ result in different $NO/NO_2$ ratios as the simulations progress.
For example, under low $NO_x$ mixing ratios the nighttime $NO/NO_2 < .001$, which means the $\delta^{15}N$
value of $NO_2$ will be close to that of total $NO_x$, which will be close to 0‰. At the same time the
$\delta^{15}N$ value of NO will be close to the fraction of the EIE achieved, which is about 50% under
low $NO_x$ conditions, resulting in a NO $\delta^{15}N$ of about -15‰. These two effects control the $\delta^{15}N$ of
$NO_2$ and that in turn controls the $\delta^{15}N$ value of $HNO_3$. In all scenarios the diurnal cycle repeats
itself over the subsequent 4 days and a greater fraction of total NO emitted has been turned into
$HNO_3$, so that by the end of the 5-day simulation the $HNO_3$ $\delta^{15}N$ values converge towards 0‰,
the defined value of $NO_x$ emissions in the simulations.

10        The modeled $\delta^{15}N$ values of HONO also have a diurnal pattern that can also be traced to
diurnal chemistry and isotope mass balance. Similar to the photolysis and photolysis + $O_3$ cases,
the HONO $\delta^{15}N$ values mirror the oscillation of the NO $\delta^{15}N$ values (data no shown). This is a
result of HONO production by the NO + OH reaction (R38). In contrast, the HONO $\delta^{15}N$ values
at night remain nearly constant despite the fact that the $\delta^{15}N$ of NO is changing dramatically.
This is because the absence of OH at night halts R38 and thus HONO production ceases and the
$\delta^{15}N$ values are simply the same as the residual daytime HONO reservoir. There is a repeated
minimum in HONO $\delta^{15}N$ values occurring each morning at 7:00 over the subsequent 4 days.
This is a result of the fact that, unlike $HNO_3$, HONO is effectively destroyed by photolysis (R4)
and OH (R45). Thus, HONO does not build up in the model over the 5-day simulation, but rather
mixing ratio peaks daily (30 ppb) at around 9:00 each day. This is when the HONO production –
destruction rate is greatest, and its mixing ratio then deceases to a low of 2 ppt by sunset. Since
the nighttime HONO, with $\delta^{15}N \sim +5.5$‰,
only contributes about 7% ($f = 0.07$) of the
morning HONO spike, it does not greatly
impact the control that NO $\delta^{15}N$ has on the
HONO $\delta^{15}N$ value. This daily isotope
effect should be contrasted with the $HNO_3$
$\delta^{15}N$ trends with time. Initially $HNO_3$
$\delta^{15}N$ values are influenced by $NO_2$ $\delta^{15}N$
variations by $NO_2$-OH-$HNO_3$ coupling,
similar to the NO-OH-HONO coupling.
But since there is no significant
photochemical sink of $HNO_3$, the control
on $HNO_3$ $\delta^{15}N$ values by $HNO_3$
accumulation increases with time, so that
by day 5 the diurnal changes in $NO_2$ $\delta^{15}N$
have almost no impact on the $HNO_3$ $\delta^{15}N$
values (Fig. 9).

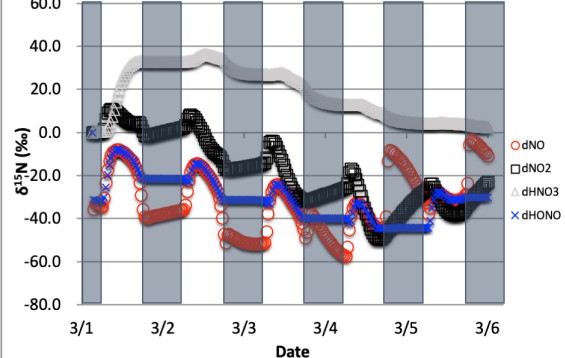

Figure 10. The time evolution of $\delta^{15}N$ values of NO, $NO_2$, $HNO_3$, and HONO caused by isotope effects of Leighton reactions, $NO_x$ isotope exchange, and $NO_2$ + OH reaction, with NO emission, simulation starts from Mar 1. The 5-day simulation was under the conditions list in Table S3c.

3.1.4 The $\delta^{15}N$ values of $NO_x$, HONO, and
$HNO_3$ due to the combined Leighton cycle, $NO_x$ isotope exchange, and $NO_2$ + OH

43        The effect of the $NO_2$ + OH reaction has on $\delta^{15}N$ values of $NO_x$ and $HNO_3$ associated
was then examined (Table S3c). Since R39 is the last step in $HNO_3$ production, the instantaneous
$\delta^{15}N\ HNO_3 = \delta^{15}N(NO_2) + \varepsilon_{48}$, thus the $\delta^{15}N$ $_{HNO3}$ is initially 40‰ higher than the $NO_2$ (Fig. 10).



This in turn depletes $^{15}N$ in the residual $NO_2$ leading to more negative $\delta^{15}N$ values in $NO_2$
relative to the Leighton + exchange simulations. These latter two effects are still in play as
evident by the diurnal $NO_x$ $\delta^{15}N$ cycling and $\Delta\delta^{15}N_{NO-NO2}$. As the 5-day simulation progresses,
the $HNO_3$ $\delta^{15}N$ value approaches 0‰, approaching the $\delta^{15}N$ of NO emissions, as expected based
on isotope mass balance. We point out that this convergence to the source $NO_x$ $\delta^{15}N$ value is
much slower in this case than the Leighton and exchanges cases. This highlights the importance
of the knowing the correct $\varepsilon_{48}$. If $\varepsilon_{48} \sim 0$ as suggested by Freyer (1993) then daytime the $\delta^{15}N$
$HNO_3 \cong \delta^{15}N$ $NO_2$, demonstrably lower than the $\varepsilon_{48} \sim 40$‰ case. In the end the average daytime
$\delta^{15}N$ value of $HNO_3$ for the entire simulation is about 10‰ higher than the $\delta^{15}N$ of the $NO_x$
source (here defined as 0‰).
## 3.2 The $\delta^{15}N$ values of $NO_x$, HONO, and $HNO_3$ due to nighttime chemistry
The role that nighttime chemistry plays in determining the $\delta^{15}N$ values of $NO_x$, HONO,
and $HNO_3$ was investigated by iteratively adding relevant fractionation factors to iRACM. The
nighttime chemistry effect was assessed by separating the effects of $NO_3$ radical chemistry and
$N_2O_5$ heterogeneous hydrolysis. $NO_3$ radical chemistry is only relevant at night because of its
short daytime lifetime with respect to photolysis, which keeps its daytime mixing ratios at the
sub $ppt_v$ levels [*Platt et al., 1984*]. At night $NO_3$ builds up and produces $HNO_3$ [*Aldener et al.,*
*2006; Finlayson-Pitts and Pitts, 1997; Horowitz et al., 1998*] via reactions with hydrocarbons
(R91-97). The magnitude of this isotope effect
was tested by adding $NO_3$ the isotope
fractionation factors for R91-97 (see methods)
and altering VOC emission rates to simulate
clean, moderate, and extreme VOC pollution
environments. Likewise, $N_2O_5$ only
accumulates at night when it begins producing
$HNO_3$ on aerosol surfaces [*Chang et al., 2011*].
The magnitude of this isotope effect was
tested by adding the $N_2O_5$ EIE (see methods)
and adding the first order $N_2O_5$ heterogeneous
pathway (see methods) to $i_NRACM$. The first-
order rate constant was adjusted to simulate
clean, polluted, and extreme pollution
environments where aerosol surface area
density largely controls the rate constant
[*Riemer et al., 2003 Chang et al., 2011*].

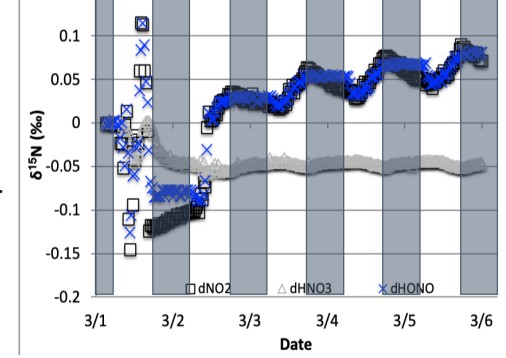

Figure 11. The difference between the $\delta^{15}N$ values of $NO_2$, HONO, and $HNO_3$ when $NO_3$ + VOC → $HNO_3$ reactions are included and excluded (NO was omitted for clarity). The 5-day simulation was under the conditions list in Table S3e. Total VOC mixing ratios during the last day of the March 1 simulation was 550-670 ppb C.

### 3.2.1 The $\delta^{15}N$ values of $NO_x$, HONO, and
$HNO_3$ due to $NO_3$ + VOC reactions
The effect on the $\delta^{15}N$ values of $NO_x$, $HNO_3$, HONO associated with the KIE occurring
during $NO_3$ + VOC nighttime reactions (R91-R97) were first examined. Four simulations were
run that included the isotope effects ($\alpha$ values in Table S6) of the Leighton cycle (R1 and R48),
$NO_x$ isotope exchange (R238), $NO_2$ + OH production of $HNO_3$ (R39), and the KIE effects (R91-





R97), as well as NO emissions. The simulation tested first was the March test case (medium VOC ~360 ppb$_v$). Then, two simulations were run for June 1 (extended sunlight, warm temperatures), one with high initial of VOC concentrations and a high VOC emission rate (2 ppb$_v$ h$^{-1}$) and one with low emission rate of VOCs (0.4 ppb$_v$ h$^{-1}$). The same two initial conditions were used in the Jan. 1 test case to assess if the extended night time and cold temperatures significantly affected the NO$_x$ of HNO$_3$ $\delta^{15}$N values produced by NO$_3$ radicals. The impact of NO$_3$ reactions on NO$_y$ $\delta^{15}$N values was determined by subtracting these simulated $\delta^{15}$N values from those same simulations when only the Leighton cycle, exchange and OH + NO$_2$ reaction was considered (Section 3.1).

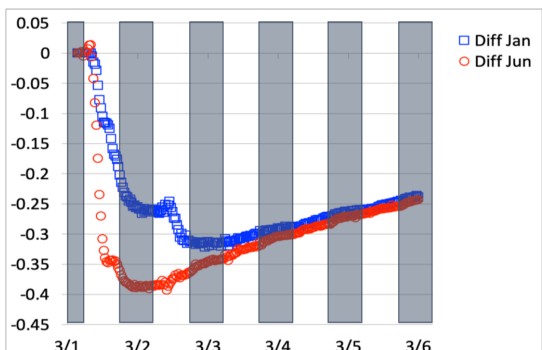

Figure 12. The difference in $\delta^{15}$N(HNO$_3$) values when NO$_3$ + VOC → HNO$_3$ reactions are included and excluded, for Mar 1 simulation, relative to Jun 1 simulation (□) and Jan 1 simulation (o). The 5-day simulation was under the conditions list in Table S3e.

The NO$_3$ + VOC KIE induced a minor diurnal pattern on the $\delta^{15}$N values of NO$_x$, and HONO, and a trend for HNO$_3$ for the March test case, but the size of the effect was relatively small (e.g., < 0.4‰; Fig. 11). At the start of the simulation (3 am) there is no HNO$_3$, therefore the initial HNO$_3$ is produced via OH production of HNO$_3$ (R39),

$\delta^{15}$N values of HNO$_3$ decreased from 0.35 to 0.2‰ during the night. The pattern is because of increasing the importance of R91-R97 in HNO$_3$ production at night. The smallness of the effect is because α values are all relatively small, the average δ for the NO$_3$ + VOC is about -4‰, and the relatively small amount of HNO$_3$ produced via these pathways (around 2.6 % of 24-hour HNO$_3$). The first source of the HNO$_3$ in the simulation (3 to 6 am) is the NO$_3$ + VOC reactions and results in a slight negative $\delta^{15}$N in HNO$_3$ value (-0.01‰). This leaves the residual NO$_3^-$ $^{15}$N enriched that is then photolyzed into NO$_2$ at sunrise and used NO$_2$ + OH → HNO$_3$ production resulting in slight positive $\delta^{15}$N values (+0.35‰) (Fig. 11). The range of the diurnal HNO$_3$ $\delta^{15}$N oscillation dampens as the fraction of emitted NO that has been converted to HNO$_3$ has increased over time. The diurnal and weekly change in $\delta^{15}$N of HNO$_3$ changes did not significantly change during the winter and summer simulations (Fig. 12) run with and without the KIE for R91-R97 show negligible differences, similar to those in Fig. 11. In conclusion, although there is some $\delta^{15}$N effect associated with NO$_3$ + VOC chemistry, it is much smaller than the effects associated with the

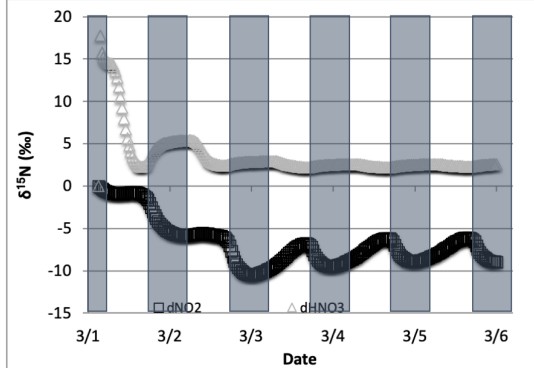

Figure 13. The difference in $\delta^{15}$N values of NO$_2$ and HNO$_3$ when the isotopic effect during N$_2$O$_5$ heterogeneous reactions (R53-54, R239) is included (α$_{N2O5}$ =1.029) and when it is excluded (α$_{N2O5}$ = 1.0). The 5-day simulation was under the conditions list in Table S3e.





Leighton cycle, $NO_2$ + OH, and $NO_x$ equilibrium.
3.2.2 The $\delta^{15}N$ values of $NO_x$, HONO, and $HNO_3$ due to $N_2O_5$ reactions
The effect on the $\delta^{15}N$ values of $NO_x$, $HNO_3$, HONO associated with the EIE of $N_2O_5$
heterogeneous hydrolysis was also tested. March 1 simulations with N emissions and $k_{N2O5} = 0.1$
$s^{-1}$ were run that included the isotope effects of the Leighton cycle (R1 and R48), $NO_x$ isotope
exchange (R238), OH production of $HNO_3$ (R39), and the $N_2O_5$ EIE (R53-54) KIE (R239)
(Table S7), as well as NO emissions. These simulations were compared to an identical
simulation but where the $\alpha_{N2O5}$ was set equal to 1.0. This ensured that the $NO_y$ chemistry was not
altered when comparing the two simulations (i.e., $\alpha_{N2O5} = 1.029$ vs. $\alpha_{N2O5} = 1.0$). The effect of
$N_2O_5$ chemistry on the $\delta^{15}N$ values of $NO_2$ and $HNO_3$ was investigated. Similar to the March 1
$NO_3$ + VOC tests, simulations with R1, R39, R48, R238, and R239 isotope effects active were
run and then compared to simulations with the same conditions but with R239 turned off. In
addition, March simulations were run using three different $k_{N2O5}$ values (.01, 0.1 and 1) and
compared to each other in order to test the range of $NO_2$ and $HNO_3$ $\delta^{15}N$ values that could be
generated solely by heterogeneous $N_2O_5$ hydrolysis.
The average daily $\delta^{15}N$ values of $HNO_3$ exhibit some diurnal oscillations that roughly
reach a steady state average value after
simulation day 2. At that point $HNO_3$ has a
$\delta^{15}N = +2.5‰$ relative to the $\alpha_{N2O5} = 1.0$
simulation. In contrast the $NO_2$ $\delta^{15}N$ values
oscillate diurnally by about +/- 2‰ around an
average daily difference of about -8‰. This
change is due to the R53-54 equilibrium,
which predicts $^{15}N$ enrichment in $N_2O_5$ (and
thus $HNO_3$) and depletion in $NO_3$ and $NO_2$.
The $N_2O_5$ produces $HNO_3$ with the highest
$\delta^{15}N$ difference (~ +29‰) during the first
simulation morning. This is because all of the
initial $HNO_3$ is produced by $N_2O_5$ due to the 3
am simulation start time. The roughly steady

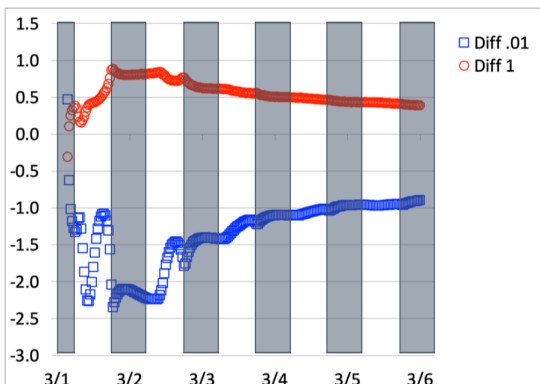

state $HNO_3$ $\delta^{15}N$ value of +2.5‰ is a
consequence of the fact that when $\alpha_{N2O5} = 1.0$

Figure 14. The difference in $\delta^{15}N(HNO_3)$ values
when the isotopic effect during $N_2O_5$
heterogeneous reactions is included and when it
is excluded, for the simulation of $k_{N2O5} = 0.1$,

$HNO_3$ is being produced by $N_2O_5$ at 0‰ and
when $\alpha_{N2O5} = 1.029$ it is being produced at

relative to 0.01 (□) and 1.0 (o). The 5-day
simulation was under the conditions list in Table
S3e.

+29‰. The ratio of this simulated +2.5‰
value and $N_2O_5$ enrichment factor of +29‰
yields 0.086, the fraction of $HNO_3$ produced by $N_2O_5$. This is similar to the fraction of $HNO_3$
produced in simulations when the $N_2O_5$ reaction was active and where it is inactive, which
yielded a fraction of 0.064. The difference in these fractions is because deactivating $N_2O_5$
chemistry changes overall $NO_y$ chemistry and $HNO_3$ production *[Dentener and Crutzen, 1990]*.
The effect of $N_2O_5$ chemistry on the $\delta^{15}N$ values of $NO_2$ is more dynamic than $HNO_3$ (Fig.
13). This is mainly due to the fact that $HNO_3$ is continually building up over time and thus its
$\delta^{15}N$ is less susceptible to change by small additions. The oscillation in the $NO_2$ $\delta^{15}N$ value





becomes more negative at night, which corresponds to the increase in the HNO$_3$ $\delta^{15}$N values. This is a reflection of $^{15}$N preferentially incorporating into N$_2$O$_5$ resulting in NO$_2$ depleted in $^{15}$N. Similar oscillations are found in NO and HONO (data not shown) as they are connected to NO$_2$ build-up and decay diurnally. This suggests that night-time partitioning of NO$_y$ will have a small but measurable influence on daytime NO$_y$ $\delta^{15}$N values. The effect of using different $k_{N2O5}$ values had a small but measurable effect on the NO$_2$ and HNO$_3$ $\delta^{15}$N values. Simulations that used a $k_{N2O5} = 1.0$ resulted in HNO$_3$ $\delta^{15}$N values that were about 2‰ lower than those run at $k_{N2O5} = 0.01$ and 1‰ heavier than when $k_{N2O5} = 1.0$. This makes sense because the mean EIE for N$_2$O$_5$ (29‰) is lower than that for NO$_2$ + OH (40‰), therefore as N$_2$O$_5$ produces more HNO$_3$ its $\delta^{15}$N value would decrease with respect to that of daytime HNO$_3$ production. Thus, the model predicts lower HNO$_3$ $\delta^{15}$N values in cold, dark polluted regions (relative to the tropics where) where N$_2$O$_5$ heterogeneous hydrolysis may be the main HNO$_3$ production pathway *[Dentener and Crutzen, 1990]*.

## 3.3 Case studies

The completed $i_N$RACM was tested using different simulation conditions that had various initial trace gas concentrations and emission rates. These test cases were labeled urban, rural, forest, and marine due to their initial conditions that were designed to mimic those environments. Initially, $i_N$RACM simulations were run using the 18 test cases without emission (9 for urban condition, 9 for rural condition), and 2 test cases with emission (1 for polluted atmosphere, 1 for clean atmosphere) provided in Stockwell *(1997)*. However, we found that trace gas concentrations in these simulations do not agree with atmospheric observations (See SI) when simulations were run for several days *[Altshuller, 1989; Baugues, 1986; Greenberg & Zimmerman, 1984; Logan, 1989; National Research Council, 1992; Torres & Buchan, 1988; Zimmerman et al., 1988]*. Thus, instead of replicating Stockwell's cases (See SI), we set up four conditions that mimic urban, rural, forest, and marine, with the initial concentrations based on various measurements from previous studies. The emission rates of NO and total VOCs were tuned until the simulation results satisfied with the following features: a). The concentration of NO$_x$ changes diurnally and stabilized through time; b). The concentration of O$_3$ changes diurnally and stabilized through time; c). VOCs are slowly consumed during nighttime (Fig. S2-5). The molar fraction of each VOC species with respect to the total VOC emission rate was obtained from Stockwell's *(1997)* emission cases. It is noted that the conditions chosen for urban, rural, forest, and marine may not be representative of all of these environments in different countries or regions (i.e all urban environments are not the same) but were devised to bracket extremes of trace gases for most tropospheric conditions (we have ignored any polar environment simulations due to isotope effects occurring during snow pack photolysis).

3.3.1 The initial concentrations and emission rates under the urban conditions

The simulations under the urban conditions were designed to simulate highly polluted urban environments such as current Asia megacities and early 1980's US cities. The initial concentration of NO$_x$ (Table S8) was set to 180 ppb, based on 90[th] percentile of polluted US cities (Cincinnati, OH, Fort Worth, TX, Memphis, TN, Miami, FL, Cleveland, OH) during summers of 1984 and 1985 *[Baugues, 1986]*, slightly lower than the maximum NO$_x$ concentration (> 200 ppb) at Shenyang, China from Aug 20 to Sept 16, 2017 *[Ma et al., 2018]*.





For urban conditions, $NO/NO_x$ ratio was set to 0.5 and $NO_x/NO_y$ ratio was set to 0.9, and we assumed the concentrations of $HNO_3$ and PAN are equal since they are minimal $NO_y$ compounds in this case. The initial carbon (C) concentration of total VOC was set to 1000 ppb C, based on the weighted average of the measurements among 30 sites during the summers of 1984 and 1985 *[Baugues, 1986]*. Baugues *(1986)* also provided the carbon fraction of toluene, xylene, HCHO, acetaldehyde, ethene, alkene other than ethene, and total alkane with respect to total VOC in ppb C. Thus, the initial concentration of toluene, xylene, HCHO, acetaldehyde, ethene was based on their carbon fraction provided by Baugues *(1986)*. The initial concentration of each alkane and alkene species was based on the carbon fraction of their groups, provided by Stockwell et al. *(1997)*. Thus, the initial concentration of total VOC was set to 252 ppb, which matches well with the recent year measurement at Shenyang, a typical urban area of Northeast China *[Ma et al., 2018]*. To stabilize the concentration of $NO_x$, the emission rates of NO and total VOCs were set to 9.36 ppb $h^{-1}$ and 7.80 ppb $h^{-1}$, respectively. The initial concentration of $O_3$ at 300 ppb, which is closed to the average among five most polluted sites from the measurement among Northeastern United States in the 1970's *[Cleveland et al., 1977]*, and the maximum value (286 ppb) measured within three main megalopolises in China (Jinjinji, Yangtze River Delta, and Pearl River Delta) from 1997 to 2016 *[Wang et al., 2017]*. During the simulations, the $O_3$ concentration stabilized, with summertime maximum hourly concentration around 200 ppb, which agree well with previous studies (National Research Council, 1992).

3.3.2 The initial concentrations and emission rates under the rural conditions

The simulations under the rural conditions were designed to simulate moderately polluted environments upwind of such current Asia megacities and early 1980's US cities. The initial concentration of $NO_x$ was set to 7 ppb (Table S8), based on the average of 120 samples collected aloft upwind of six US cities (Dallas-Fort Worth, TX, Tulsa, OK, Birmingham, AL, Atlanta, GA, Philadelphia, PA, New York, NY) during summers of 1985 and 1986 *[Altshuller, 1989]*. The 7 ppb $NO_x$ also matches well with the value measured in Baltimore/Washington airshed in 2011 *[He et al., 2013]*. For rural condition, $NO_x/NO_y$ ratio was set to 0.44, based on the average ratio among previous studies *[Carroll et al., 1992; Fahey et al., 1986; National Research Council, 1992; Parrish et al., 1986; Williams et al., 1987]*, $NO/NO_x$ ratio was set to 0.7. According to Logan *(1989)*, the ratio between $NO_x$ and PAN has a median value of around 2.5. Thus, the initial concentrations of NO, $NO_2$, $HNO_3$, and PAN are 4.9 ppb, 2.1 ppb, 6 ppb, and 3 ppb, respectively (Table S8). The initial concentrations of each VOC species were based on the weighted average of the measurements, upwind of the six US cities. The 8.3 ppb initial concentration of total OC matches well with the recent year measurement in the rural area of Midwest, US *[Sjostedt et al., 2011]*. The initial concentration of $O_3$ was set to 50 ppb, based on the average summertime $O_3$ concentration among rural areas *[Cooper et al., 2012, Janach, 1989; Logan, 1989]*. To stabilize the concentration of $NO_x$ and $O_3$, the emission rates of NO and total VOCs were set to 0.24 ppb $h^{-1}$and 0.59 ppb $h^{-1}$, respectively.

3.3.3 The initial concentrations and emission rates under the forest conditions

The simulations under the forest conditions were designed to simulate low polluted environments (Table S8), with high emissions of biogenic VOCs such as isoprene. The initial concentration of $NO_x$ was set to 0.06 ppb, which is the minimum concentration at the central basin of Amazon tropical forest *[Torres and Buchan, 1988]*. Typically, at the remote sites (forest and marine condition), the ratio of $NO_x/NO_y$ is between 0.1 and 0.2, $HNO_3/NO_x$ is less than 5,





and PAN/NO$_x$ is less than 1 *[Carroll et al., 1992]*. To satisfy all three conditions, the initial concentration of HNO$_3$ and PAN were set to 0.29 ppb and 0.05 ppb, respectively. NO/NO$_x$ ratio was set to 0.9. Thus, NO and NO$_2$ equal to 0.054 ppb and 0.006 ppb, respectively. The initial concentrations of each VOC species were based on the median concentration of 81 samples in Amazon *[Zimmerman et al., 1988]*. The 6.7 ppb initial concentration of total VOC matches well with the recent year measurement in Amazon *[Fuentes et al., 2016]*. The initial concentration of O$_3$ was set to 10 ppb, which is closed to the average hourly concentration during the summertime in Amazonia *[Fuentes et al., 2016, Kirchhoff, 1988]*. To stabilize the concentration of NO$_x$ and O$_3$, the emission rates of NO and total VOCs were set to 9.4 ppt h$^{-1}$ and 0.35 ppb h$^{-1}$, respectively.

### 3.3.4 The initial concentrations and emission rates under the marine conditions

The simulations under the marine conditions were designed to simulate remote, clean oceanic environments. The initial concentration of NO$_x$ was set to 0.03 ppb (Table S8), which is the medium value of 28,974 data points, measured simultaneously at Mauna Loa, Hawaii, from May 1 to June 4, 1988 *[Carroll et al., 1992]*, as well as at Cape Norman, Canada, northern Atlantic coastal site, from February to April, 1996 *[Yang et al., 2004]*. Typically, at the remote sites (forest and marine condition), the ratio of NO$_x$/NO$_y$ is between 0.1 and 0.2, HNO$_3$/NO$_x$ is less than 5, and PAN/NO$_x$ is less than 1 *[Carroll et al., 1992]*. To satisfy all three conditions, the initial concentration of HNO$_3$ and PAN were set to 0.145 ppb and 0.025 ppb, respectively. NO/NO$_x$ ratio was set to 0.9. Thus, NO and NO$_2$ equal to 0.027 ppb and 0.003 ppb, respectively. The initial concentrations of each VOC species were based on the median values between the sample collected by two cruises over the tropical Pacific Ocean, during December 1982, and July 1982, respectively *[Greenberg and Zimmerman, 1984]*. The 5.4 ppb initial concentration of total VOC matches well with the recent year measurement over the western North Pacific and eastern Indian Ocean *[Saito et al., 2000]*. The initial concentration of O$_3$ was set to 10 ppb, which the average summertime concentration among maritime sites in both the north and south hemispheres *[Janach, 1989]*. To stabilize the concentration of NO$_x$ and O$_3$, the emission rates of NO and total VOCs were set to 12.48 ppt h$^{-1}$ and 0.234 ppb h$^{-1}$, respectively

### 3.3.5 The simulated $\delta^{15}$N value under the urban conditions

The simulated $\delta^{15}$N values of NO$_x$ under the urban conditions have extreme diurnal oscillations and almost no weekly trend. The most striking feature is the extreme diurnal oscillations in NO and NO$_2$ $\delta^{15}$N values. During the early nighttime (1/2, 6/2) the NO and NO$_2$ $\delta^{15}$N values are similar (~ -5‰, $\Delta\delta^{15}$N$_{NO-NO2}$ =+20 ‰), but they diverge over about a 6 hour period until reaching a maximum $\Delta\delta^{15}$N$_{NO-NO2}$ = -35 +/- 5%. At this point, the NO $\delta^{15}$N value has reached its minimum values of -38‰ (Jan 1) and -36‰ (June 1), which corresponds to minimums in the $f_{NO}$ and NO mixing ratio (SI Fig x). Meanwhile, the $\delta^{15}$N values of NO$_2$ approach 0‰ as its mixing ratio and $f_{NO2}$ have reached maximums. This indicates that the NO$_x$ isotopic exchange ($\varepsilon_{NO/NO2}$ = -35‰ at 298K) is dominating the NO$_x$ isotopic effects during the night and that under these conditions it requires about 6 hours for NO$_x$ to achieve full isotopic equilibrium.



In contrast, during the daytime, the
NO and $NO_2$ $\delta^{15}N$ values are influenced
by photolysis and Leighton cycle isotope
effects. In the morning, the split between
NO and $NO_2$ $\delta^{15}N$ values is eliminated
as $NO_2$ is photolyzed. This is, in large
part, a simple isotope mass balance
effect because isotopically heavy
(nighttime) $NO_2$ is now producing
isotopically heavy NO causing the NO
$\delta^{15}N$ values to increase, closely
following the $f_{NO}$ (SI Fig. x). The $NO_2$
PHIFE ($\varepsilon_{R1a}$ = 2.3‰ at 45 degree of
solar zenith angle), however, must also
be in play otherwise $NO_2$ $\delta^{15}N$ values
would not decrease in the early morning.
The $NO_2$ PHIFE, however, cannot
account for the roughly 20‰ decrease in
the $NO_2$ $\delta^{15}N$ values over the course of
the mid-morning. This extra decrease is
the result of $NO_2$ + OH ($\varepsilon_{39a}$ = 40‰)
which depletes the residual $NO_2$ $\delta^{15}N$
values and results in an $HNO_3$ $\delta^{15}N$
increase, which maximizes around noon
each day (Fig. 15). This $\varepsilon_{39a}$ effect
reduces in importance in the late

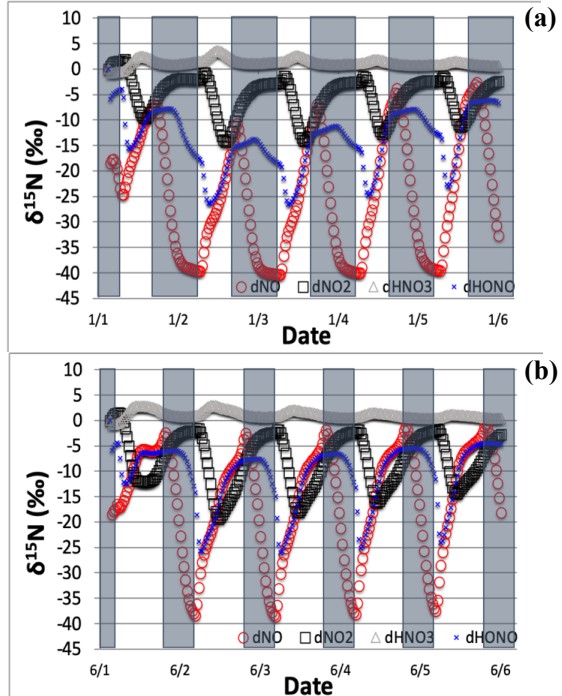

Figure 15. The $\delta^{15}N$ values of NO, $NO_2$, $HNO_3$, and HONO for urban condition for Jan 1 (top, a) and Jun 1 (bottom, b) simulation. The 5-day simulation was under the conditions list in Table S8.

afternoon as OH mixing ratios decrease and the rate of R39 decreases to zero. The $NO_x$ $\delta^{15}N$
equivalence point ($NO_2$ $\delta^{15}N$ ≈ NO $\delta^{15}N$) occurs around noon in June and early evening in
January, and corresponds to the $NO_2$ $\delta^{15}N$ minimum values of ~ -15 to -20 ‰. This highlights
the fact that actinic flux plays a key role in controlling the rate of $NO_x$ $\delta^{15}N$ oscillation rates. The
combined Leighton cycle isotope effects eventually push NO $\delta^{15}N$ values above the $NO_2$ values,
which occurs sooner in the June simulation than the Jan. because of higher $NO_x$ Leighton cycle
turnovers caused by higher photolysis rates and $O_3$ mixing ratios (maximums 316 $ppb_v$ versus
236 $ppb_v$). Thus, unlike nighttime, when $NO_x$ isotope exchange controls $NO_x$ $\delta^{15}N$ values,
photolysis and Leighton cycle and R39 isotope effects control daytime $NO_x$ $\delta^{15}N$ values that in
turn control HONO and $HNO_3$ $\delta^{15}N$ values.
The $\delta^{15}N$ values of HONO mimic the $\delta^{15}N$ values of NO during the daytime because of the
production by the NO + OH reaction (R38). This mimicking is most obvious in the non-urban
simulations where daytime HONO and NO $\delta^{15}N$ values are the same during the day, afterward,
the HONO $\delta^{15}N$ decouples and remains constant during the night when OH is absent (Fig. 16-18).
Interestingly, unlike the non-urban simulations, the urban nighttime HONO $\delta^{15}N$ does not remain
constant. There is a slight increase in HONO $\delta^{15}N$ values earlier during the late evening, then the
$\delta^{15}N$ values of HONO decrease dramatically to approach the $\delta^{15}N$ values of NO after midnight
(Fig. 15). Since no HONO isotope effects included in the current $i_N$RACM, this nighttime effect
was traced to nighttime OH production under high pollution conditions. During the daytime, OH
is produced by the reaction of water vapor with $O^1D$, which arises from ozone photolysis. In
contrast, during the night, due to the absence of photolysis, the production of OH is usually
assumed to be zero. However, under high VOCs, the $i_N$RACM model simulates OH production
through a 2-step process. First, production of $HO_2$ by VOC + $NO_3$ reactions (R91, R93, R96)
followed by the production of OH by $HO_2$ + $NO_x$ reactions (R41-R44). Under the urban
condition, the relatively high VOC concentration promotes the production of $HO_2$ by VOC +
$NO_3$, and relatively high ozone concentration leads to higher $NO_2$ and $NO_3$ concentration during
the early evening, resulting in the production of OH by $HO_2$ + $NO_x$ reactions (R41-R44).
Because of the production of HONO by the NO + OH reaction (R38), the $\delta^{15}N$ values of HONO
starts approaching the nighttime $\delta^{15}N$ values of NO, when the concentration of OH becomes
sufficiently high. This effect can be seen in the changing OH mixing ratios during the urban
night (SI Fig. X).
The daily $HNO_3$ $\delta^{15}N$ values reach a daily maximum around noon each simulation day and
trend toward 0‰ at night and by the end of the week. The initial $HNO_3$ $\delta^{15}N$ values are near 0‰
because the initial $HNO_3$ is set to 0‰ (Table S8). Afterward, the daily $HNO_3$ $\delta^{15}N$ values reach
midday maximums, ~4‰ for Jan 1 and 3‰ for June 1 simulations, which corresponds to the
maximums in OH concentration and $HNO_3$ produced by the R38 pathway. The $\delta^{15}N$ values of
$HNO_3$ decrease during the late afternoons and nighttime, as the $\delta^{15}N$ of the reacting $NO_2$
decreases and as isotope effects of $NO_3$ +VOCs and $N_2O_5$ become effective. During this 5-day
simulation, the $HNO_3$ concentrations gradually reach quasi-equilibrium, as an increasing amount
of $NO_x$ is converted into $HNO_3$. As a result,
the small diurnal cycle in $HNO_3$ $\delta^{15}N$
values becomes less obvious going from
simulation day 1 to day 5 where it
approaches 0‰, the default $\delta^{15}N$ of $NO_x$
emissions, which obeys the N isotope mass
balance. In June 1 simulation, the rate and
duration of ozone photolysis is higher, thus
more $O^3P$ is produced, comparing to Jan 1
simulation. As a result, the concentration of
$NO_3$ during the late afternoon is higher,
causing the isotope effect of $NO_3$ +VOCs
reaction being stronger. Therefore, the $\delta^{15}N$
values of $HNO_3$ in June 1 simulation
reaches the maximum value and approaches
to 0‰ faster than in Jan 1 simulation.
**3.3.6 The simulated $\delta^{15}N$ value under the**
**rural conditions**
The simulated $\delta^{15}N$ values of $NO_x$
under the rural conditions shows similar,
yet strikingly different, diurnal and weekly
trends compared to urban simulation. This
difference is mainly due to the longer $NO_x$
isotope equilibrium timescale under lower
$NO_x$ conditions. The $\Delta\delta^{15}N_{NO-NO2}$ decreases

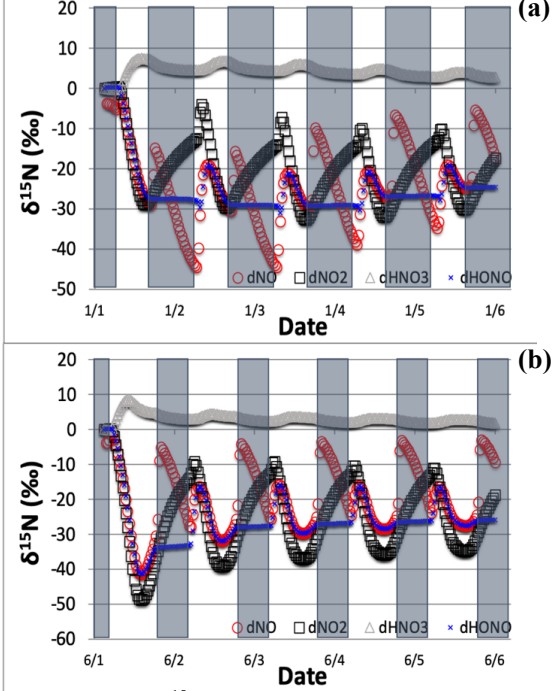

Figure 16. The $\delta^{15}N$ values of NO, $NO_2$, $HNO_3$, and HONO for rural condition for Jan 1 (top, a) and Jun 1 (bottom, b) simulation. The 5-day simulation was under the conditions list in Table S8.





during the nighttime and reaches the -15.3‰ (June 1) just before sunrise indicating that $NO_x$
isotopic exchange ($\varepsilon_{NO/NO2}$ = -40‰ at 298K) is not reaching full equilibration during the night.

3       Similar to urban simulation, $\Delta\delta^{15}N_{NO-NO2}$ in Jan 1 simulation is lower than in June 1
simulation, caused by more remaining amount of $NO_x$ after weaker $NO_2$ photolysis during the
daytime. The $NO_x$ isotopic exchange is weaker, comparing to the urban simulation, due to the
lower $NO_x$ concentrations. Therefore, the decrease of $\Delta\delta^{15}N_{NO-NO2}$ during the nighttime is less
obvious than urban simulation. The $\Delta\delta^{15}N_{NO-NO2}$ gradually increases during the daytime, as the
photolysis ($\varepsilon_{R1a}$ = 2.3‰ at 45 SZA), $O_3$ + NO ($\varepsilon_{48a}$ = -6.7‰), and $NO_2$ + OH ($\varepsilon_{39a}$ = 40‰)
isotope effects become effective. The maximum daytime $\Delta\delta^{15}N_{NO-NO2}$ reaches at sunset, showing
5.5‰ for Jan 1 simulation and 8.5‰ for June 1 simulation. The $\Delta\delta^{15}N_{NO-NO2}$ increases by 37.6‰
and 23.8‰ during the daytime for Jan 1 and June 1 simulation, respectively, which indicates the
dominance of $NO_2$ + OH reaction (R39). Compared to urban simulation, change of $\Delta\delta^{15}N_{NO-NO2}$
during the daytime is smaller, caused by weaker isotope effect from $O_3$ + NO (R48) reaction,
because of lower ozone concentration. The change of $\Delta\delta^{15}N_{NO-NO2}$ during the daytime for June 1
simulation is larger than that for Jan 1 simulation, due to the higher rate and longer duration of
$NO_2$ photolysis.

17       Similar to the urban simulation, the $\delta^{15}N$ values of HONO mimics the $\delta^{15}N$ values of NO
during the daytime because of the production by the NO + OH reaction (R38), of which the
fractionation factor ($\alpha$) is zero. Unlike urban simulation, however, the $\delta^{15}N$ values of HONO
remain constant at night, since the
concentration of $HO_x$ (OH + $HO_2$) is not
enough to produce HONO, due to the
relatively low VOC (3.3% of urban value)
and ozone (16.7% of urban value)
concentration. A key caveat about the
$\delta^{15}N$ values of HONO is that we have
excluded any KIE associated with the NO
+ OH reaction because it has not been
measured or calculated. Since this is the
termination reaction, any isotope effect in
this reaction would have a large influence
on HONO $\delta^{15}N$ values. In the forest and
ocean environment simulations, the $\delta^{15}N$
values of HONO also mimic the $\delta^{15}N$
values of daytime NO so they will not be
discussed in the subsequent sections of
this paper.

38       The $\delta^{15}N$ values of $HNO_3$ increases
during the mornings and reaches the
maximum of 7.5‰ for Jan 1 simulation
and 8.4‰ for June 1 simulation, as the
isotope effects of $NO_2$ + OH ($\varepsilon_{39a}$ = 40‰)
greater than $NO_3$ +VOCs ($\Sigma\varepsilon_{91a\sim97a}$ = -
27.8‰). The maximum $\delta^{15}N$ value of
$HNO_3$ is higher than that in urban

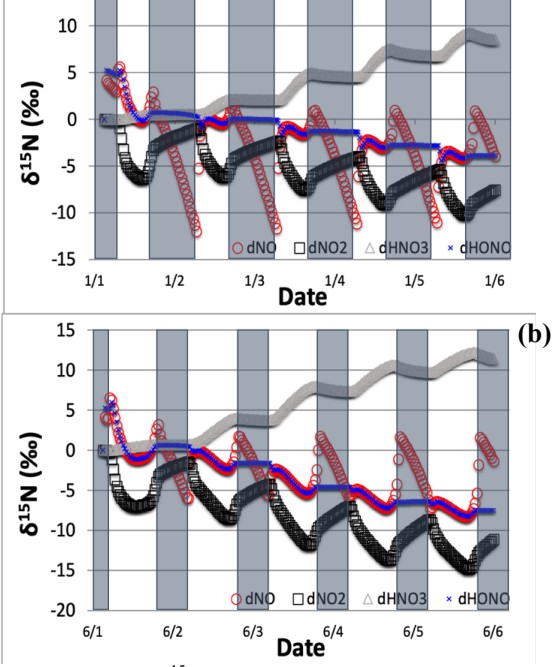

Figure 17. The $\delta^{15}N$ values of NO, $NO_2$, $HNO_3$, and HONO for forest condition for Jan 1 (top, a) and June 1 (bottom, b) simulation. The 5-day simulation was under the conditions list in Table S8.





simulation, as the lower VOC concentration weakens the $NO_3$ +VOCs reactions. The $\delta^{15}N$ values
of $HNO_3$ decreases during the late afternoons and nighttime, as the isotope effects of $NO_3$
+VOCs are effective. The same as urban simulation, the diurnal trend of $\delta^{15}N(HNO_3)$ becomes
less obvious from simulation day 1 to day 5 and approaches 0‰ at the end of the simulation, as
an increasing amount of $NO_x$ being converted into $HNO_3$. Again, the $\delta^{15}N$ values of $HNO_3$ in
June 1 simulation approaches to 0‰ faster than in Jan 1 simulation, due to more $O(^3P)$ from
stronger $O_3$ photolysis.

### 3.3.7 The simulated $\delta^{15}N$ value under the forest conditions

The simulation of $\delta^{15}N$ values of $NO_x$ under the forest conditions shows similar, yet
significantly different diurnal and weekly trend, comparing to urban and rural simulation. There
are obvious differences between the daily oscillations in $NO_2$ $\delta^{15}N$ values in the forest relative to
urban simulations. In the cleaner atmosphere, $NO_2$ $\delta^{15}N$ values only change daily by only about
5‰, whereas they oscillate by 30 to 40‰ in the urban case. Likewise, daily oscillations in $NO$
$\delta^{15}N$ values are weaker in the forest conditions (5-10‰) than in the urban and suburban
conditions (30-40‰). This results in a decrease in the $\Delta\delta^{15}N_{NO-NO2}$ during the nighttime,
reaching minimum values of -11.0‰ (Jan 1) and -4.5‰ (June 1) compared to roughly -40‰ in
the urban case. This shows that under low $NO_x$ conditions the $NO_x$ isotopic exchange cannot
occur fast enough to reach its full effect. This effect is more pronounced in June, due to the short
night, relative to the January simulations that have about 14 hours of darkness at this latitude.
This is similar to urban and rural simulations, where $\Delta\delta^{15}N_{NO-NO2}$ in Jan 1 simulation is lower
than in June 1 simulation, because of reduced $NO_2$ photolysis hours. The decrease of $\Delta\delta^{15}N_{NO-NO2}$ during the nighttime in the forest conditions is less obvious than urban and rural simulation.
The $\Delta\delta^{15}N_{NO-NO2}$ gradually increases during the daytime, as the photolysis ($\varepsilon_{R1a}$ = 4.2‰), $O_3$ +
$NO$ ($\varepsilon_{48a}$ = -6.7‰), and $NO_2$ + OH ($\varepsilon_{39a}$ = 40‰) isotope effects become effective. This results in
daytime $NO$ $\delta^{15}N$ values that are less negative than those in $NO_2$, opposite of the urban case
where $NO_2$ $\delta^{15}N$ values are either higher or equal to $NO$. As a consequence, the daytime
$\Delta\delta^{15}N_{NO-NO2}$ values are positive (as opposed to negative in the urban case) and reach a maximum
at sunset (6.5‰ for Jan 1, 7.1‰ for June 1). The $\Delta\delta^{15}N_{NO-NO2}$ increases by 17.5‰ and 13.3‰
during the daytime for Jan 1 and June 1 simulation, respectively, which indicates the dominance
of $NO_2$ + OH reaction (R39). The change of $\Delta\delta^{15}N_{NO-NO2}$ during the daytime is smaller than both
urban and rural simulation, due to lower ozone concentration. The change of $\Delta\delta^{15}N_{NO-NO2}$ during
the daytime for June 1 simulation is larger than that for Jan 1 simulation, due to the higher rate
and longer duration of $NO_2$ photolysis.

The most striking difference between the "clean" and "polluted" simulations is the
separation of $HNO_3$ $\delta^{15}N$ values from the initial (emission) $NO_x$ $\delta^{15}N$ value (defined as 0‰). The
$\delta^{15}N$ values of $HNO_3$ increases daily by about 1-4‰ due to the isotope effects of $NO_2$ + OH ($\varepsilon_{39a}$
= 40‰) but are constant or slightly decrease throughout the night due to $NO_3$ +VOCs reactions
under these conditions. This leads to a stepwise increase in $HNO_3$ $\delta^{15}N$ values that reach
maximums of about +10‰ by the end of the 5-day simulation. This is in contrast to the urban
and suburban simulations where $HNO_3$ $\delta^{15}N$ values reach minimums (~ 0‰) at the end of the
simulation. This is an isotope mass balance effect driven by how N is partitioned into $NO_y$ under
different conditions. Under high $NO_x$ and VOC conditions (urban, rural) over 90% of emitted
$NO_x$ has portioned into $HNO_3$ by the end of the 5-day simulation, thus the $HNO_3$ $\delta^{15}N$ value
approaches that of the $NO_x$ emissions. In contrast, under low $NO_x$ and VOC conditions (forest,



ocean) only about 33% of emitted $NO_x$ has portioned into $HNO_3$, with the bulk of remainder as $NO_x$ (21%) and organic nitrate (42%) and PAN (4%). In this case, the isotope effects incorporated into the reactions that are responsible for this partitioning manifest themselves in the $\delta^{15}N$ of the individual $NO_y$ compounds. For example, $i_N$RACM predicts that forest conditions will produce $\delta^{15}N$ values in organic nitrate even though there are no isotope effects associated with organic nitrate production or loss in $i_N$RACM. This highlights how the $\delta^{15}N$ values are tracing shifts in $NO_y$ oxidation pathways.

### 3.3.8 The simulated $\delta^{15}N$ value under the marine conditions

The simulation of $\delta^{15}N$ values of $NO_y$ under the marine conditions is very similar to the forest simulation. The daily oscillations and weekly change in the $\delta^{15}N$ of $NO_x$, HONO and $HNO_3$ all follow the same pattern as the forest simulations but with slight amplification in all compounds. The nighttime change in NO $\delta^{15}N$ is 15‰ (Jan) and 10‰ (June) is about 5‰ larger than in the forest simulation. Similarly, the ocean condition nighttime change in $NO_2$ $\delta^{15}N$ (9 to 12‰) is about 5‰ larger than in the forest simulation. The result is that the $\Delta\delta^{15}N_{NO-NO2}$ values decrease during the nighttime, reaching minimum values of -13.5‰ (Jan 1) and -3.6‰ (June 1). Again this shows that under low $NO_x$ conditions the $NO_x$ isotopic exchange is not occurring fast enough to reach its full effect ($\varepsilon_{NO/NO2}$ = -35‰ at 298K) and is more pronounced in the summer months due to the short night., relative to the January simulations that have about 14 hours of darkness at this latitude. Similar to the forest simulation the $\delta^{15}N$ values of $HNO_3$ increase stepwise

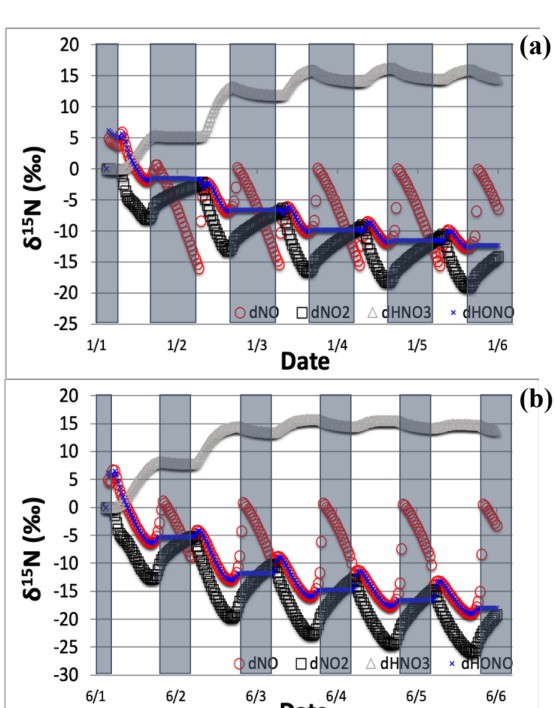

Figure 18. The $\delta^{15}N$ values of NO, $NO_2$, $HNO_3$, and HONO for marine condition for Jan 1 (top, a) and Jun 1 (bottom, b) simulation. The 5-day simulation was under the conditions list in Table S8.

during the daytime and reach the maximum of 16‰ for both the Jan and June simulations. This is about 5‰ larger than in the forest simulation and 15‰ higher than the urban case. Similar to forest simulation, the emission rate of $NO_x$ is higher than the conversion rate of $NO_x$ to $HNO_3$. As a result, the amount of $NO_x$ increases from simulation day 1 to day 5. The abundant $NO_x$ promotes the production of $HNO_3$ during the daytime by $NO_2$ + OH reaction (R39), which leads to the overall increasing trend throughout the simulation period.

### 3.4. Model comparison with observations

There are a number of challenges when trying to compare the $i_N$RACM model predictions of $NO_y$ $\delta^{15}N$ values with observations in real world. First, there has yet to be a study where the



$\delta^{15}N$ values of NO, $NO_2$, and $NO_3^-$ have been simultaneously measured. The most abundant data
is on the $\delta^{15}N$ value of $NO_3^-$ in aerosols or rainwater. Even with these studies, a direct
comparison is difficult because of the $\delta^{15}N$ value of the source $NO_x$ may be variable in space and
time. The $\delta^{15}N$ value of $NO_x$ sources can range from -40 to + 20 ‰ and both $NO_x$ sources and
$NO_3^-$ deposition will be a strong function of the transport history of the air mass that is sampled.
Without a 3-D chemical transport model that includes the $i_N$RACM mechanism, a direct
comparison with most $NO_3^-$ $\delta^{15}N$ studies would be tenuous. In addition, most $NO_y$ $\delta^{15}N$ studies
provide neither trace gas concentrations ($NO_x$, $O_3$, CO, VOC) nor local trace gas emissions that
would be required to constrain $i_N$RACM for it make an accurate prediction of secondary
pollutants or $\delta^{15}N$ values.

11       The most complete dataset for which to evaluate the $i_N$RACM mechanism is from Riha

(2013) in a study in Tucson AZ, USA. In that
study $PM_{2.5}$ and $PM_{10}$ were collected weekly
(24-hour period) for one year (2006) and the
$\delta^{15}N$ value of water soluble $NO_3^-$ was
determined (Figure 1). Into PM mass and
$NO_3^-$ $\delta^{15}N$ data, local measurements of traces
gases (accept VOCs) and meteorology
(temperature, relative humidity, wind) were
available. In addition, detailed local primary
pollutant emission inventories have been
developed (Diem and Comrie, 2001). Tucson
is a city with little industry or power
generation so roughly 80% of the $NO_x$ is due
to vehicles and the relative proportion of all
NOx sources is invariant throughout the year.
Further, Tucson is surrounded by a desert
landscape and by and large not influences by
regional pollution sources outside the city.
These factors overcome some of the
uncertainties discussed above. $i_N$RACM was
initialized with observed trace gas
concentrations and $NO_x$ and VOC emissions
were based on previous work (Riha, 2013) and
the source $NO_x$ $\delta^{15}N$ value was set to -3‰,
typical of vehicle emissions (Walter et al.,
2015) and run on the first day of each month.
The predicted $NO_3^-$ (as $HNO_3$) $\delta^{15}N$ values
(After 48 hours) matched remarkably well
with the observed values in $PM_{2.5}$ and $PM_{10}$
(Figure 19). Observed maximums were in the
winter months, peaking January at 15‰ close
to the model maximum in January of 17%.
The minimum $\delta^{15}N$ values (-2‰) are
measured in July, similar to model predictions
of 0‰ during July. The model captures the

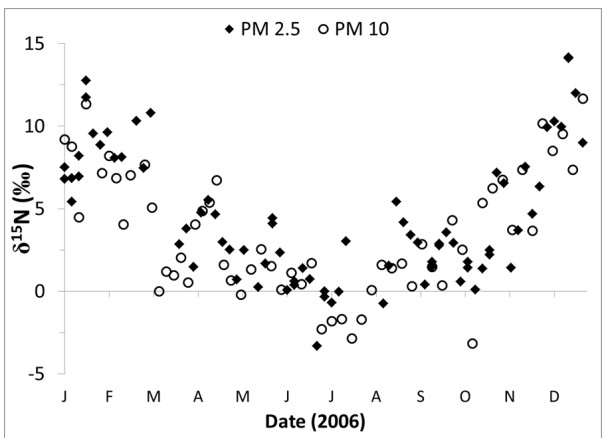

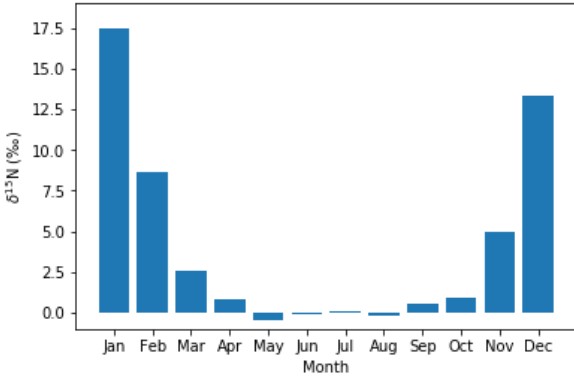

Figure 19. Upper panel is the observed $NO_3^-$ $\delta^{15}N$ values of PM in the city of Tucson (Riha, 2013). Lower panel is the $NO_3^-$ $\delta^{15}N$ values of $HNO_3$ predicted by the $i_N$RACM mechanism. Minimums, maximums, and seasonal change in $\delta^{15}N$ in PM $NO_3^-$ can be explained by the EIE, KIE, and PHIFE occurring during $NO_y$ cycling.





seasonal trend quite well, including the Spring plateau.   This suggests that at this location, the
observed seasonal variation in PM $NO_3^-$ $\delta^{15}N$ values can be explained isotope effects associated
with the photochemical conversion of NOx into $HNO_3$.
## 4. Conclusion

6       We have developed the first 0-D photochemical box model for $^{15}N$ compounds in the
tropospheric $NO_x$-$NO_y$ cycle.   It was shown that of the 100's of N reactions in the RACM
mechanism only a handful significantly impact the main $NO_y$ compounds ($NO_x$, HONO, $HNO_3$).
Primarily these are Leighton cycle reactions, $NO_2 + OH$, and $NO_x$ isotope exchange, with $N_2O_5$
and nitrate radical reactions having a significant, but minor influence on $NO_y$ $\delta^{15}N$ values.   It was
also shown that there were two factors that can dramatically influence the simulated $NO_y$ $\delta^{15}N$
values.   The first is the size of the isotope fractions factors (KIE, EIE, PHIFE) for any given
reaction.   For example, the large EIE (assumed) for $NO_2 + OH$ was much more important than
the small KIE associated with $NO_3 + VOC$ reactions. This highlights the need for direct or
computational measurements of KIE, EIE, PHIFE in $NO_y$ reactions, particularly R39. The
second is that shifts in oxidation pathways caused by pollutant loading are being reflected in the
$NO_y$ $\delta^{15}N$ values.   In particular, high $NO_x + VOC$ environments with aerosols tend to favor $\delta^{15}N$
that reflects $NO_x$ isotope exchange and $N_2O_5$ uptake, while clean environments favor $\delta^{15}N$ that
reflects $NO_x$ cycle and OH oxidation reactions. This highlights that $NO_y$ $\delta^{15}N$ values are not only
related to $NO_x$ sources but also affected by $NO_y$ chemistry.
The $i_N$RACM model makes a number of predictions that could be tested by measuring the
$\delta^{15}N$ values of various $NO_y$ compounds in different environments and at different temporal
scales. First, the model predicts very large diurnal changes in $NO_x$ $\delta^{15}N$ values in all
environments, ranging from 10 to 40‰, which could be easily be detected with even the crudest
isotope methods (± 2‰).   Second, it predicts that in highly polluted environments the $\delta^{15}N$ value
of $HNO_3$ will be close to the $\delta^{15}N$ value of the $NO_x$ sources in the area, but in clean
environments, it will be 10 to 15‰ heavier. Third, it predicts seasonal and latitudinal trends in
$HNO_3$ $\delta^{15}N$ values driven by sunlight and the shifting photochemical pathways associated with it.
It predicts higher winter $HNO_3$ $\delta^{15}N$ values, as $NO_x$ isotope exchange becomes more important
relative Leighton and OH reactions that become dominant in the summer.   This effect should be
more pronounced as a function of latitude. There should be relatively minor changes in
equatorial $HNO_3$ $\delta^{15}N$ values since sunlight hours do not vary, significant changes at mid-
latitudes (50% seasonal sunlight change) change, and essentially a bimodal change at the poles
(ignoring snowpack recycling effects). Fourth, it predicts there will be $\delta^{15}N$ variations in key
$NO_y$ reservoirs yet to be measured such as organic nitrates and PAN. Finally, the $i_N$RACM
model predicts that the most dramatic changes in $NO_y$ $\delta^{15}N$ changes will occur after rain events
where $NO_y$ is largely removed from the atmosphere by wet depositions. Post rain, the $NO_y$ $\delta^{15}N$
values effectively "reset", particularly $HNO_3$, and will have their biggest difference relative to
the $NO_x$ $\delta^{15}N$ before trending to the $NO_x$ source over time. The $i_N$RACM model suggests that
knowing how is $NO_y$ partitioned and the $\delta^{15}N$ value of one (or more) compound that the $\delta^{15}N$ of
the $NO_x$ source can be determined. This, in turn, can be used as a constraint on $NO_x$ budgets
from the local to regional and global scale.
This effect that tropospheric photochemistry has on $NO_y$ $\delta^{15}N$ values was tested and
shown to general initially lead to higher $\delta^{15}N$ values in $HNO_3$ relative to the initial $NO_x$. The
difference between the $\delta^{15}N$ of $HNO_3$ relative to the initial (emitted) $NO_x$ was typically ~ +10‰





by the 2nd and 3rd day of the simulation. This seems consistent with observations that show $NO_3^-$ $\delta^{15}N$ values (positive $\delta^{15}N$) are typically higher than most $NO_x$ sources (negative $\delta^{15}N$). This difference between $NO_x$ source and $HNO_3$ $\delta^{15}N$ values tends to diminish as the simulation progresses as either all of the initial $NO_x$ is oxidized to $HNO_3$ (no emission simulations) or the proportion of $HNO_3$ to total N approaches 1 (emission scenarios). This type of bias can be eliminated by incorporating $i_N$RACM into 3-D chemical transport models that account for time-dependent deposition and emission of $NO_y$.

The model accuracy and its validation could be improved with additional research. The $i_N$RACM model could be refined by additional theoretical and/or experimental determination of the isotope fractionation factors for the N reactions. First and foremost the fractionation factor for the $NO_2$ + OH reaction needs evaluating in a more robust manner. Likewise, the fractionation factor for the NO + OH, another 3-body reaction, will have a large influence on HONO $\delta^{15}N$ values and determining its value will be key for interesting future HONO $\delta^{15}N$ data. The fractionation factor for $NO_2$ photolysis requires attention given the limitation of the $\Delta$ZPE PHIFE model [*Blake et al.*, *2003; Liang et al., 2004; Miller and Yung, 2000*]. On the validation end, the simultaneous measurement of $\delta^{15}N$ in multiple $NO_y$ compounds would expose the accuracy or limitations of the $i_N$RACM model in a quantitative way. Repeating these simultaneous measurements in a range of environments would test the predictions made by out test case simulations.

**Code availability**: Fortran code and associated input files are archived on Zenodo.org https://zenodo.org/. DOI/10.5281/zenodo.3834920. An online version of this $i_N$RACM model is available for public use at https://mygeohub.org/tools/sbox/

**Author contribution**: Greg Michalski was the lead investigator for the project designed the modeling experiments, organized the tasks, and wrote the manuscript. Huan Fang and David Mase modified the RACM code to include $^{15}N$ isotopes, assisted in writing and editing the manuscript. Wendell W Walters derived EIE, KIE, and PHIFE used in the model and assisted in writing and editing the paper

**Acknowledgements:** We would like to thank the Purdue Research Foundation and the Purdue Climate Change Research Center for providing funding for the project. We would like to thank Bo Sun for helping with the Fortran coding and data generation.

The authors declare that they have no conflict of interest.





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
