# Peer review of "iNRACM: Incorporating 15N into the Regional Atmospheric Chemistry Mechanism (RACM) for assessing the role photochemistry plays in controlling the isotopic composition of NOx, NOy, and atmospheric nitrate."

_Geoscientific Model Development, 2020_

## Referee Comment (RC1) · Lei Geng (Referee) · 8 Aug 2020

This manuscript by Michalsk et al provides a framework that calculates the differences in the atmospheric transformation of different N-15 substituted NOy molecules, i.e., the isotope fractionations caused by the chemical conversions of NOx to different final products. Despite some weakness in the parameterizations of the fractionation factors which are needed to quantify the reaction rate constant of N-15 substituted molecules, the framework serves as a tool to test the sensitivity of, e.g., d15N of atmospheric HNO3, to the NO-NO2 cycling and the subsequent conversion pathways of NO2 to HNO3. This will be very helpful to understand the variations of atmospheric HNO3 measured in different environments. Nevertheless, I feel the manuscript needs some more illustrating data/figures to better understand/evaluate the ability of the model as well as the model outputs. In addition, I think the manuscript needs a better proofread, as there are apparently some typo or mislabeling of figure or equation numbers (I will specify later).

My major concerns/comments are as follows:

1. Most of the reaction rate constants of the N-15 substituted molecules in the model were estimated by multiplying the general chemical reaction rate constant by the fractionation factor. This technologically is not a big issue, however, in some specific reactions, I think the authors oversimplified the treatments. Especially, for the case of OH + NO2→ HNO3 reaction. The authors considered this is an EIE instead of KIE, as it is a third body reaction that contains two steps: firstly formed an activated *HNO3, and which can be deactivated to HNO3 or decompose to NO2 and OH. The authors stated the first step, OH + NO2 <-> *HNO3 is the dominant isotope step and that induces a large fractionation factor. But I don't quite get the point: if this is a EIE process, then during the lifetime of *HNO3 to HNO3, would an isotope equilibrium between NO2 and *HNO3 be able to fully established? The author should compare the rate of *HNO3 decomposition, and its rate of collision with M to form HNO3, if the decomposition rate is orders of magnitude larger than the collision rate to form HNO3, then the assumption of OH+NO2 → HNO3 is an EIE with alpha of 40 permil is valid. Otherwise, the isotope effect caused by this reaction need to be reevaluated, as well as the entire model results as currently in the model this reaction accounts a large fraction of the overall isotope difference from NOx to HNO3.

2. The treatment of N2O5 hydrolysis: The model doesn't contain aerosol chemistry but considering the importance of N2O5 hydrolysis in nighttime HNO3 production, this has to be involved in the model. The authors treated this as a first order reaction to go directly as $k*[N2O5] \rightarrow 2HNO3$. This is a common method, but in order to get it right (so the isotopes are correct), the authors at least need to compare the production of HNO3 from this parametrized first-order reaction is realistic, i.e., it will dominate nighttime HNO3 production and account for a large fraction of total nitrate production in the system. This is because N2O5 hydrolysis is a dominant HNO3 production pathway, contributing to ~ 40 % of total HNO3 budget which comparable with the OH + NO2 pathway in the daytime (Alexander et al., 2019 ACP). Unfortunately, from the limited figures in SI, it appears the nighttime HNO3 production is often flattened in the four simulation cases. If this is true, then the parametrization of the N2O5 reaction in the model is probably wrong. P.s., this is one of the reasons that I think more illustrating data/figure needs to be provided. To evaluate the isotope behaves in the model, the chemistry has to be first correctly simulated. So the authors need to first show the day and night HNO3 production and discuss whether the production close enough to the observations.

3. Mass balance. As a box model, it is a closed system, so mass, as well as isotopes has to be conserved. Otherwise there might be something wrong with the model setup. So I think the authors should show a mass balance figure, with the time evaluation of total N, as well as total N-15 in the model. This is important to evaluate the overall model performance.

4. I am curious why the model predict a large seasonal variations in d15N of HNO3 for the case of Tucson city, but in the four case simulations (i.e., urban, rural, forest and marine) there were minimum diurnal cycle in d15N of HNO3 especially in the last few days of the simulations. This is strange, as explained by the authors, the relative importance of NO isotope exchange versus Leighton cycle and OH reaction determined the seasonal cycle, but at the same time, won't be the shift of this relative importance from day to night larger than that in the seasonal scale? So why the modeled diurnal cycle is so small compared to the modeled seasonal cycle?  In addition, the authors state the Tucson case d15N HNO3 was reported as 48 hours simulation result, I wondered what it will look like

if for 4 or 5 days simulation, which is the typical lifetime of atmospheric nitrate. In other words, why picked 48 hours?

5. During the day time, the model result indicates that NO-NO2 isotope exchange is very small compared to the Leighton cycle and the OH reaction. I just wanted to see more data to prove this, i.e., can the authors compare the rate of exchange versus the reaction rates of the Leighton cycle and OH reaction during the daytime?

6. The last, I suggest to authors to add another case simulation, that in the middle of 5 day simulation, varying the d15N of emitted NO from 0 to, e.g., 10 permil, and see how the isotopes of NOy in the system vary. This will be interesting as in real environment, NOx emitted from different sources and its isotope vary all the time.

The following are some general comments:

1. P6, line 38-39, this approach has also been mentioned by Bao et al. 2015 (https://doi.org/10.1016/j.gca.2015.07.038) and He et al. 2020 (https://bg.copernicus.org/preprints/bg-2020-120/).

2. Section2.3.1, for the sensitivity tests in this and other similar sections, it is unclear how many chemistry are involved. For example, in figure 2, was nighttime chemistry involved? If yes, why d15N of HNO3 stays the same at night but when d15N of NO2 is very low?

3. Section 2.3.1, the second paragraph, the discussion on daytime NO3 and NO2O5, I think this can be made less complicated to just show the mass of these molecules during the daytime, their negligible mass during daytime is the reason of their negligible isotope effects.

4. Section 3.14, what is eplison-48? If you meant the NO2 + Oh reaction, won't it be e-39?

5. Section 3.3.5, line 35: .d15N(NO-NO2) = + 20 ‰, but From Figure 15, I didn't see this 20 per mil difference in early nighttime of 1/2 and 6/2;

6. Same section, line 37: Shouldn't be Jan. 2 and Jun 2?

7. Same section, last sentence: "conditions it requires about 6 hours for NOx to achieve full isotopic equilibrium", I doubt this. First, Walters et al. paper in GRL 2016 actually shows the exchange is fast. In addition, in this simulations, it seems the d15N difference between NO and NO2 reach the maximum by late night, but this could be a result of mass

balance, i.e., when almost all NO is converted to NO2, and d15N of NO2 approaches to zero (the starting value), and by mass balance NO will be very negative. It will be much more helpful to understand this if the fraction of NO overnight can be plotted. This issue also exists for all other case simulations. In addition, the model has continued NO emissions with d15N of zero (not a completely closed system), how does this continued add-up of NO affect the isotopes of the system?

8. P28, Line 35-36: can you pull out the rate of isotope exchange between NO and NO2, as well as the rate of photolysis and Leighton cycle at daytime and night? it is surprising that at daytime isotope exchange appears to be negligible.

9. P29, Line 11-12, what is SI Fig. X? and there is no OH figure in SI.

---

## Referee Comment (RC2) · Anonymous Referee #2 · 20 Aug 2020

Comment on "i$_N$RACM: Incorporating $^{15}$N into the Regional Atmospheric Chemistry Mechanism (RACM) for assessing the role photochemistry plays in controlling the isotopic composition of NO$_x$, NO$_y$, and atmospheric nitrate" by Michalski et al.

**General Comments:**

Michalski and coauthors have expanded an existing chemical mechanism designed for air pollution chemistry to consider the heavy isotope of nitrogen ($^{15}$N) as it proceeds through the NO$_x$ and NO$_y$ cycles. Although they have made significant assumptions due to lack of experimental data, they have demonstrated the feasibility of using the tool to simulate idealized and real-world cases. They finally make the argument that incorporating this scheme in a 3-D air quality model will enable researchers to use $\delta^{15}$N data to constrain NO$_x$ emissions, a major challenge facing the air pollution community. Broadly, I think the elements of the study are exceptionally well-designed and chosen, particularly the progression from simplified box model examples to more complicated scenarios and finally one real-world case. I also thank the authors for a well-scoped and executed introduction and methods that helped me understand much better the pertinent issues with and potential gains from a better understanding of NO$_y$ isotope abundance. This may be one of those frustrating reviews that reveal the ignorance and misunderstandings of the reviewer, but hopefully the authors will find these comments useful for the success of the paper. My suggestions involve mostly matters of presentation.

The manuscript is overlong and could stand another read-through focused on eliminating redundant text and moving details to the SI. Section 2.4 for example, could be moved. I think upgrading section 2.4.4 out of section 2.4 is warranted. This section could also describe the Tuscon case.

Figures 1, 4, 5 can be moved to the SI.

A table is needed in the main text to summarize the scenarios in sections 3.1 and 3.2. It is too difficult to track what reactions are included or excluded from the different cases and quite frustrating to interpret axes like Fig. 12 where two different cases are subtracted. Fig 12 and Fig. 14 require y-axis labels.

Another suggestion includes moving sections 3.3.1-3.3.4 to the SI and instead using an abbreviated table to report the initial conditions and emissions of each scenario, with only the most important compounds from Table S8 (e.g. NO, NO2, HNO3, PAN, (and isotopes) and total VOC).

Regarding the case in section 3.4, I agree with the other anonymous reviewer that 48 hours may be problematic as it appears to be within the dynamic phase of the 5-day simulation examples in section 3.3. It would be prudent to run the model out to 5 days or longer to get a sense for the steady-state that will be reached.

How were the emissions for the Tuscon case applied? What temporal assumptions were made? What is the sensitivity to this?

How are the assumptions with respect to the heterogenous reaction of N2O5 impacting the results in Fig. 19? What is the expected impact from developments to this part of the mechanism in the future?

Are there significant concentrations of $^{15}NH_4$ and $^{15}NH_3$ in the atmosphere? If so, can this impact the measured $\delta^{15}N$ data for $PM_{2.5}$ and $PM_{10}$?

**Typos/Editorial Suggestions**

1. Page 2, Line 18: "The final mechanism was **characterized**…**"**
2. Page 6, Line 26-28: I was confused about exactly how the alpha relates to the reaction shown. Is it isotope product over isotope reactant? Could you explain a bit more in the text please? Just an extra five words or so may do it.
3. Page 6, Line 28: I can't find this alpha in the SI. R238 and R238A have different alphas than this.
4. Table S2b: This is a difficult lift at this stage of the manuscript, but it would have been helpful if R238 were named R238b instead.
5. Page 11, Lines 30-34: I was confused by this assertion and the use of the terms *chemically equilibrated* vs *isotopically equilibrated*. The authors assert that HNO3 is likely not able to exchange isotopes in the gas phase, but N2O5 and NO3 are? How is HNO3 not already chemically equilibrated in the mechanism as well, for example in R239a and R239b? I would have been better-served with a more precise definition of equilibration and
6. Page 27. Line 33: Recommend avoiding "weekly trend" throughout discussion and instead using something like "multi-day" trend. I don't think there's enough data here to establish weekly trends.

---

## Author Comment (AC1) · 30 Mar 2021

We would like to thank the two reviewers for their thoughtful and constructive comments and appolgize for the delayed response. We struggled for some time with how to shorten the manuscript (reviewer 1) while at the same time answering questions raised by reviewer. It is always a trade-off between clarity and brevity. We were following

Stockwell's original RACM paper format (model, mechanisms, test cases) to have a mirror comparison. We then realized that realized that 3.3 section is not absolutely relevant since there is no observational data to evaluate of each the test cases and that section 3.3 would be better as a stand-alone paper. There we can dig deeper into the issue brought up by the reviewers and include other effects (sunlight, emissions). Also it would allow us to make predictions that could be tested by observations, it is too bad we didn't think of that earlier! Therefore, we have cut 3.3 ( 8 pages), sections 2.4.4 and other text and reduced paper's length by 9 pages. This also resolves the many issues of the below by both reviewers. Thank the reviewers for guiding us to this change of course. Below we address the other reviewer's concerns.

The manuscript is overlong and could stand another read-through focused on eliminating redundant text and moving details to the SI. Section 2.4 for example, could be moved. I think upgrading section 2.4.4 out of section 2.4 is warranted. This section could also describe the Tuscon case.

No one knows about this article length better than the authors! See first paragraph

Figures 1, 4, 5 can be moved to the SI. A table is needed in the main text to summarize the scenarios in sections 3.1 and 3.2. It is too difficult to track what reactions are included or excluded from the different cases and quite frustrating to interpret axes like Fig. 12 where two different cases are subtracted. Fig 12 and Fig. 14 require y-axis labels.

See reply to comment 1.

Another suggestion includes moving sections 3.3.1-3.3.4 to the SI and instead using an abbreviated table to report the initial conditions and emissions of each scenario, with only the most important compounds from Table S8 (e.g. NO, NO2, HNO3, PAN, (and isotopes) and total VOC). See reply to comment 1

Regarding the case in section 3.4, I agree with the other anonymous reviewer that 48

hours may be problematic as it appears to be within the dynamic phase of the 5-day simulation examples in section 3.3. It would be prudent to run the model out to 5 days or longer to get a sense for the steady-state that will be reached. The samples are 24 hour PM collections. We based our 48 hour simulation based on an estimated lifetime of the airmass at the collections. At an average wind speed of 2 m/s and a 20 km diameter of the city (and urban island in the middle of the desert), air mass lifetimes are on the order of 3-4 hours. 48 is well beyond that accounting for possible stagnant air. The unfortunate limitation of the box model is that at true equilibrium, the HNO3 '15N will simply equal the NOx '15N. These are other issues we will address in the follow up paper.

How were the emissions for the Tuscon case applied? What temporal assumptions were made? What is the sensitivity to this? We assume constant monthly annual emissions throughout the year and day. This is the advantage of the Tucson site is the lack of industry and significant vegetation and that the emissions are essentially constant (mainly vehicles), we show this in a new supplement figure from a student thesis. Hourly emissions are another matter, but this is irrelevant given the sampling time of NO3- to which we are comparing (24 hr). The Stockwell model is not easily amendable to hourly changes in emissions. The second paper will give us the opportunity to explore this effect.

How are the assumptions with respect to the heterogenous reaction of N2O5 impacting the results in Fig. 19? This is small, the we have added another supplement showing daily PM mass for the year, which is 40 ug/m3 +/- 10 ug. Since the N2O5 fraction (10 %) is small to begin with and the N2O5 fractionation is not widely different than OH pathway the variation from N2O5 is small. Another issue we will explore in the follow up paper.

What is the expected impact from developments to this part of the mechanism in the future? We have embellished on this in the conclusion section

Are there significant concentrations of 15NH4 and 15NH3 in the atmosphere? If so, can this impact the measured $\delta$15N data for PM2.5 and PM10?

The analysis technique for determining the 15N of NO3- is specific to NO3- only (and NO2-) My major concerns/comments are as follows: 1. Most of the reaction rate constants of the N-15 substituted molecules in the model were estimated by multiplying the general chemical reaction rate constant by the fractionation factor. This technologically is not a big issue, however, in some specific reactions, I think the authors oversimplified the treatments. Especially, for the case of OH + NO2aÌĂHNO3 reaction. The authors considered this is an EIE instead of KIE, as it is a third body reaction that contains two steps: firstly formed an activated *HNO3, and which can be deactivated to HNO3 or decompose to NO2 and OH. The authors stated the first step, OH + NO2 <-> *HNO3 is the dominant isotope step and that induces a large fractionation factor. But I don't quite get the point: if this is a EIE process, then during the lifetime of *HNO3 to HNO3, would an isotope equilibrium between NO2 and *HNO3 be able to fully established? The author should compare the rate of *HNO3 decomposition, and its rate of collision with M to form HNO3, if the decomposition rate is orders of magnitude larger than the collision rate to form HNO3, then the assumption of OH+NO2aÌĂHNO3 is an EIE with alpha of 40 permil is valid. Otherwise, the isotope effect caused by this reaction need to be reevaluated, as well as the entire model results as currently in the model this reaction accounts a large fraction of the overall isotope difference from NOx to HNO3. We agree that the NO2 + OH fractionation factor is uncertain. We can estimate this based on approximate lifetime of the activated complex and the reaction rate. The bimolecular collisional rate constant for NO2 + OH →NO2OH* is ∼ is of 2 x 10-10 compared to the recently updated measured high pressure rate constant 6.3 x 10-11. This means formation and decomposition of the activated complex is 3 times faster than HNO3 production sufficient time for equilibrium of N isotopologues since they only require a single collision and separation to achieve equilibrium (unlike O atoms). Conceptually one can think of an ensemble of 1 x 106 NO2 molecules, with natural abundance isotopologues = air N2. The two isotopolugues collide with OH kf at

approximately the same rate, and disassociate kr by of calculated factor of 1.04 (0.96) then the activated complex ensemble will have isotopically equilibrated by this single cycle. We will explore this effect, and all sensitivity to all fractionation factors, in detail in the subsequent paper 2. The treatment of N2O5 hydrolysis: The model doesn't contain aerosol chemistry but considering the importance of N2O5 hydrolysis in nighttime HNO3 production, this has to be involved in the model. The authors treated this as a first order reaction to go directly as k*[N2O5] aÌÃ 2HNO3. This is a common method, but in order to get it right (so the isotopes are correct), the authors at least need to compare the production of HNO3 from this parametrized first-order reaction is realistic, i.e., it will dominate nighttime HNO3 production and account for a large fraction of total nitrate production in the system. This is because N2O5 hydrolysis is a dominant HNO3 production pathway, contributing to $\sim$ 40 % of total HNO3 budget which comparable with the OH + NO2 pathway in the daytime (Alexander et al., 2019 ACP). Unfortunately, from the limited figures in SI, it appears the nighttime HNO3 production is often flattened in the four simulation cases. If this is true, then the parametrization of the N2O5 reaction in the model is probably wrong. P.s., this is one of the reasons that I think more illustrating data/figure needs to be provided. To evaluate the isotope behaves in the model, the chemistry has to be first correctly simulated. So the authors need to first show the day and night HNO3 production and discuss whether the production close enough to the observations. As noted in the text the first order absolute rate constants are based on the current literature values, which are a function of aerosol surface area. Our plausible range of first order rate constants is based on these experiment/observational studies, therefore they cannot be "wrong" in the model unless the body of peer review literature is incorrect. Figure 13. Our kN2O5 = 1.0 s-1 is simalr to others such as Yvon et al. (1996, kN2O5 = 1.0 s-1), Riemer et al. ( kN2O5 = 0.9 s-1). It is difficult to access the effect in simulations because much is happening, this was the point of figures 13 and 14. 3. Mass balance. As a box model, it is a closed system, so mass, as well as isotopes has to be conserved. Otherwise there might be something wrong with the model setup. So I think the authors should show a mass balance figure,

with the time evaluation of total N, as well as total N-15 in the model. This is important to evaluate the overall model performance. We agree but the figure would simply be a straight line of zero so aa figure would not be enlightening. We have added "These simulations were also used to test whether iNRACM achieve N isotope mass balance via S15N/S14N where the sums are the ending abundances of all N compounds. This resulted in d15N = 0 for all simulations." to page 8

4. I am curious why the model predict a large seasonal variations in d15N of HNO3 for the case of Tucson city, but in the four case simulations (i.e., urban, rural, forest and marine) there were minimum diurnal cycle in d15N of HNO3 especially in the last few days of the simulations. This is strange, as explained by the authors, the relative importance of NO isotope exchange versus Leighton cycle and OH reaction determined the seasonal cycle, but at the same time, won't be the shift of this relative importance from day to night larger than that in the seasonal scale? So why the modeled diurnal cycle is so small compared to the modeled seasonal cycle? In addition, the authors state the Tucson case d15N HNO3 was reported as 48 hours simulation result, I wondered what it will look like if for 4 or 5 days simulation, which is the typical lifetime of atmospheric nitrate. In other words, why picked 48 hours? Two things are happening during these simulations. The first is near instantaneous isotope fractionation during the photochemistry and the second is isotope mass balance. Similar to the simulations, the real world atmosphere will reflect these two competing processes. At the beginning of the simulations the isotopic change in secondary N compounds (NOy except NOx) is large because there is no initial concentration of these compounds and their resulting '15N is entirely due to the isotope fractionation factors. As the simulation progresses those NOy compounds that are stable and build up, HNO3 in particular will always approach the '15N of the NO emission by mass balance, ie by the end of the five day simulation 99% of the emitted NO has been converted to HNO3 and thus must approximately have the same '15N as the NO because is a closed system. In the real world the system is open, either due to deposition (wet/dry removal of HNO3) or advective transport. Our 48 hour choice was based on the size of the

city and a low average windspeed which would replace the regional air mass at least every 48 hours. A 5 day simulation is imposing a stagnant 5 day air mass over the city, which is unrealistic. The model highlight the importance of atmospheric lifetimes in controlling '15N of NOy. At very short lifetimes (post rainy day) the '15N should partition strongly, but with long lifetimes the values are controlled by mass balance, in particular long lived reservoirs like HNO3 or PAN. The annual trend is driven by hours of daylight. Something we will explore in detail in subsequent paper. 5. During the day time, the model result indicates that NO-NO2 isotope exchange is very small compared to the Leighton cycle and the OH reaction. I just wanted to see more data to prove this, i.e., can the authors compare the rate of exchange versus the reaction rates of the Leighton cycle and OH reaction during the daytime? The daytime results show NOx exchange lite is similar to or less than Leighton, but is condition dependent. Both these lifetimes are on the order of 100 sec during peak sunlight and moderate NOx mixing rations (few 10s ppb). The lifetime of NO2 with respect to OH + NO2 is several hours at peak sunlight. The question is the exchange relative to Leighton which will be a function of NOx mixing ratio and changing j coefficients. Evaluating this is the purpose of figure 8 (in original). Those show at high NOx exchange is still dominant even during the daytime (8a), but Leighton becomes dominant at low NOx during day, and slowly achieve equilibrium at night (8c) At steady state -NO2/dt = NO2/dt = 1/jNO2 = 1/(.001s) = 100 s -dNO2/dt = HNO3/dt = 1/k[OH] = 1/(6.3*10-11)(2.5 x106) = 6350 s 6. The last, I suggest to authors to add another case simulation, that in the middle of 5 day simulation, varying the d15N of emitted NO from 0 to, e.g., 10 permil, and see how the isotopes of NOy in the system vary. This will be interesting as in real environment, NOx emitted from different sources and its isotope vary all the time. This would be interesting, but we avoid this simulation for two reasons. The first is that the entire objective of the paper is to show potential '15N changes in the absence of any '15N "sources effect". This objective is laid out on page 4, lines 34-42. Adding source variation would add layers of complexity and add confusion. Second the emissions in this box model are at a fixed rate (and ratio) and would require reprogramming the model to change on a $\frac{1}{2}$ hour basis (time step) and there is not much current evidence on how these ratio emissions would values change hourly in the real world. This paper we explicitly say that this is only evaluating the photochemical effect on 15N . We have new papers in review that address the source effect, and another that will assess source (and mixing) and chemistry combined. The following are some general comments: 1. P6, line 38-39, this approach has also been mentioned by Bao et al. 2015 (https://doi.org/10.1016/j.gca.2015.07.038) and He et al. 2020 (https://bg.copernicus.org/preprints/bg-2020-120/). The quantum approach goes all the way back to Urey, Mayer, and Biegelisen! and is common knowledge in the field 2. Section 2.3.1, for the sensitivity tests in this and other similar sections, it is unclear how many chemistry are involved. For example, in figure 2, was nighttime chemistry involved? If yes, why d15N of HNO3 stays the same at night but when d15N of NO2 is very low? We are not fully understanding the reviewer's question. In every sensitivity test all N reactions are replicated and all a=1 except for one reaction. This is repeated for each N reaction. For each of those 96 simulations we test if NOx, HONO, or HNO3 15N changes by 1 permil or more. Fig 1 (1) shows a non-sensative reaction (NO3 + NO) and Fig. 2 shows a sensitive reaction (NO2→NO + O). How 15N is partitioned the way they do is a complex function initial concentrations, emission rates, simulation length, isotope mass balance. . .etc. These are discussed in the subsequent sections. 3. Section 2.3.1, the second paragraph, the discussion on daytime NO3 and NO2O5, I think this can be made less complicated to just show the mass of these molecules during the daytime, their negligible mass during daytime is the reason of their negligible isotope effects. We agree that NO3, N2O5 is low at night, but this is not the intent of the section. The intent was to show that "sensitive/nonsensative" classification of the reactions was only valid using the NOx, HNO3, HONO as the test molecules. The intent of this section is to show that reactions producing/consuming NO3 and N2O5 could impact their 15N values (ie. NO3 photolysis) but not necessarily the NOx,HNO3, and HONO. This effect would be independent of their concentration. The point being that IF you wanted to model NO3 15N then one would need the more exotic fractionation factors, but not for the main compounds. 4. Section 3.14, what is eplison-48? If you meant the NO2 + Oh reaction, won't it be e-39? Yes, this was a typo! 5. Section 3.3.5, line 35: .d15N(NO-NO2) = + 20 ‰ but From Figure 15, I didn't see this 20 per mil difference in early nighttime of 1/2 and 6/2; 6. Same section, line 37: Shouldn't be Jan. 2 and Jun 2? 7. Same section, last sentence: "conditions it requires about 6 hours for NOx to achieve full isotopic equilibrium", I doubt this. First, Walters et al. paper in GRL 2016 actually shows the exchange is fast. In addition, in this simulations, it seems the d15N difference between NO and NO2 reach the maximum by late night, but this could be a result of mass balance, i.e., when almost all NO is converted to NO2, and d15N of NO2 approaches to zero (the starting value), and by mass balance NO will be very negative. It will be much more helpful to understand this if the fraction of NO overnight can be plotted. This issue also exists for all other case simulations. In addition, the model has continued NO emissions with d15N of zero (not a completely closed system), how does this continued add-up of NO affect the isotopes of the system? 8. P28, Line 35-36: can you pull out the rate of isotope exchange between NO and NO2, as well as the rate of photolysis and Leighton cycle at daytime and night? it is surprising that at daytime isotope exchange appears to be negligible. 9. P29, Line 11-12, what is SI Fig. X? and there is no OH figure in SI. These are all explainable, and will be addressed per comment reply 1.

---

## Author Response (AR1)

We would like to thank the two reviewers for their thoughtful and constructive comments. We struggled for quite some time with how to shorten the manuscript (reviewer 1) while at the same time answering questions raised by reviewer. It is always a trade-off between clarity and brevity. We were following Stockwell's original RACM paper format (model, mechanisms, test cases) to have a mirror comparison. We then realized that realized that  3.3 section, the test cases is not absolutely relevant since there is no observational data to evaluate of each the test cases and that section 3.3  would be better as a stand-alone paper.  There we can dig deeper into the issue brought up by the reviewers and include other effects (sunlight, emissions). Also it would allow us to make predictions that could be tested by observations, it is too bad we didn't think of that earlier! Therefore, we have cut 3.3 ( 8 pages), sections 2.4.4 and other text and reduced paper's length by 9 pages. This also resolves the many issues of the below by both reviewers. Thank the reviewers for guiding us to this change of course. Below we address the other reviewer's concerns.

The manuscript is overlong and could stand another read-through focused on eliminating redundant text and moving details to the SI. Section 2.4 for example, could be moved. I think upgrading section 2.4.4 out of section 2.4 is warranted. This section could also describe the Tuscon case.

No one knows about this article length better than the authors!  See first paragraph

Figures 1, 4, 5 can be moved to the SI. A table is needed in the main text to summarize the scenarios in sections 3.1 and 3.2. It is too difficult to track what reactions are included or excluded from the different cases and quite frustrating to interpret axes like Fig. 12 where two different cases are subtracted. Fig 12 and Fig. 14 require y-axis labels.

See reply to comment 1.

Another suggestion includes moving sections 3.3.1-3.3.4 to the SI and instead using an abbreviated table to report the initial conditions and emissions of each scenario, with only the most important compounds from Table S8 (e.g. NO, NO2, HNO3, PAN, (and isotopes) and total VOC).
See reply to comment 1

Regarding the case in section 3.4, I agree with the other anonymous reviewer that 48 hours may be problematic as it appears to be within the dynamic phase of the 5-day simulation examples in section 3.3. It would be prudent to run the model out to 5 days or longer to get a sense for the steady-state that will be reached.
The samples are 24 hour PM collections.  We based our 48 hour simulation based on an estimated lifetime of the airmass at the collections.  At an average wind speed of 2 m/s and a 20 km diameter of the city (and urban island in the middle of the desert), air mass lifetimes are on the order of 3-4 hours. 48 is well beyond that accounting for possible stagnant air. The unfortunate limitation of the box model is that at true equilibrium, the $HNO_3$  $\delta^{15}N$ will simply equal the NOx $\delta^{15}N$. These are other issues we will address in the follow up paper.

How were the emissions for the Tuscon case applied? What temporal assumptions were made? What is the sensitivity to this?

We assume constant monthly annual emissions throughout the year and day. This is the advantage of the Tucson site is the lack of industry and significant vegetation and that the emissions are essentially constant (mainly vehicles), we show this in a new supplement figure from a student thesis. Hourly emissions are another matter, but this is irrelevant given the sampling time of $NO_3^-$ to which we are comparing (24 hr). The Stockwell model is not easily amendable to hourly changes in emissions. The second paper will give us the opportunity to explore this effect.

How are the assumptions with respect to the heterogenous reaction of N2O5 impacting the results in Fig. 19?

This is small, the we have added another supplement showing daily PM mass for the year, which is 40 ug/m3 +/- 10 ug. Since the $N_2O_5$ fraction (10 %) is small to begin with and the $N_2O_5$ fractionation is not widely different than OH pathway the variation from $N_2O_5$ is small. Another issue we will explore in the follow up paper.

What is the expected impact from developments to this part of the mechanism in the future?
We have embellished on this in the conclusion section

Are there significant concentrations of $^{15}NH_4$ and $^{15}NH_3$ in the atmosphere? If so, can this impact the measured $\delta^{15}N$ data for PM2.5 and PM10?

The analysis technique for determining the $\delta^{15}N$ of $NO_3^-$ is specific to $NO_3^-$ only (and $NO_2^-$)

My major concerns/comments are as follows:
*1. Most of the reaction rate constants of the N-15 substituted molecules in the model were estimated by multiplying the general chemical reaction rate constant by the fractionation factor. This technologically is not a big issue, however, in some specific reactions, I think the authors oversimplified the treatments. Especially, for the case of OH + NO2àHNO3 reaction. The authors considered this is an EIE instead of KIE, as it is a third body reaction that contains two steps: firstly formed an activated *HNO3, and which can be deactivated to HNO3 or decompose to NO2 and OH. The authors stated the first step, OH + NO2 <-> *HNO3 is the dominant isotope step and that induces a large fractionation factor. But I don't quite get the point: if this is a EIE process, then during the lifetime of *HNO3 to HNO3, would an isotope equilibrium between NO2 and *HNO3 be able to fully established? The author should compare the rate of *HNO3 decomposition, and its rate of collision with M to form HNO3, if the decomposition rate is orders of magnitude larger than the collision rate to form HNO3, then the assumption of OH+NO2àHNO3 is an EIE with alpha of 40 permil is valid. Otherwise, the isotope effect caused by this reaction need to be reevaluated, as well as the entire model results as currently in the model this reaction accounts a large fraction of the overall isotope difference from NOx to HNO3.*

We agree that the $NO_2$ + OH fractionation factor is uncertain. We can estimate this based on approximate lifetime of the activated complex and the reaction rate. The bimolecular collisional rate constant for NO2 + OH →NO2OH* is ~ is of 2 x 10^{-10} compared to the recently updated

measured high pressure rate constant 6.3 x 10$^{-11}$. This means formation and decomposition of the activated complex is 5 times faster than HNO$_3$ production sufficient time for equilibrium of N isotopologues since they only require a single collision and separation to achieve equilibrium (unlike O atoms). Conceptually one can think of an ensemble of 1 x 10$^6$ NO$_2$ molecules, with natural abundance isotopologues = air N2. The two isotopolugues collide with OH k$_f$ at approximately the same rate, and disassociate k$_r$ by of calculated factor of 1.04 (0.96) then the activated complex ensemble will have isotopically equilibrated by this single cycle.    We will explore this effect, and all sensitivity to all fractionation factors, in detail in the subsequent paper

2. *The treatment of N2O5 hydrolysis: The model doesn't contain aerosol chemistry but considering the importance of N2O5 hydrolysis in nighttime HNO3 production, this has to be involved in the model. The authors treated this as a first order reaction to go directly as k\*[N2O5] à 2HNO3. This is a common method, but in order to get it right (so the isotopes are correct), the authors at least need to compare the production of HNO3 from this parametrized first-order reaction is realistic, i.e., it will dominate nighttime HNO3 production and account for a large fraction of total nitrate production in the system. This is because N2O5 hydrolysis is a dominant HNO3 production pathway, contributing to ~ 40 % of total HNO3 budget which comparable with the OH + NO2 pathway in the daytime (Alexander et al., 2019 ACP). Unfortunately, from the limited figures in SI, it appears the nighttime HNO3 production is often flattened in the four simulation cases. If this is true, then the parametrization of the N2O5 reaction in the model is probably wrong. P.s., this is one of the reasons that I think more illustrating data/figure needs to be provided. To evaluate the isotope behaves in the model, the chemistry has to be first correctly simulated. So the authors need to first show the day and night HNO3 production and discuss whether the production close enough to the observations.*

As noted in the text the first order absolute rate constants are based on the current literature values, which are a function of aerosol surface area. Our plausible range of first order rate constants is based on these experiment/observational studies, therefore they cannot be "wrong" in the model unless the body of peer review literature is incorrect. Figure 13. Our k$_{N2O5}$ = 1.0 s-1 is simialr to others such as Yvon et al. (1996, k$_{N2O5}$ = 1.0 s$^{-1}$), Riemer et al. ( k$_{N2O5}$ = 0.9 s-1). It is difficult to access the effect in simulations because much is happening, this was the point of figures 13 and 14.

3. *Mass balance. As a box model, it is a closed system, so mass, as well as isotopes has to be conserved. Otherwise there might be something wrong with the model setup. So I think the authors should show a mass balance figure, with the time evaluation of total N, as well as total N-15 in the model. This is important to evaluate the overall model performance.*

We agree but the figure would simply be a straight line of zero so aa figure would not be enlightening. We have added "These simulations were also used to test whether i$_N$RACM achieve N isotope mass balance via S$^{15}$N/S$^{14}$N where the sums are the ending abundances of all N compounds. This resulted in d$^{15}$N = 0 for all simulations." to page 8

4. I am curious why the model predict a large seasonal variations in d15N of HNO3 for the case of Tucson city, but in the four case simulations (i.e., urban, rural, forest and marine) there were minimum diurnal cycle in d15N of HNO3 especially in the last few days of the simulations. This is strange, as explained by the authors, the relative importance of NO isotope exchange versus Leighton cycle and OH reaction determined the seasonal cycle, but at the same time, won't be the shift of this relative importance from day to night larger than that in the seasonal scale? So why the modeled diurnal cycle is so small compared to the modeled seasonal cycle? In addition, the authors state the Tucson case d15N HNO3 was reported as 48 hours simulation result, I wondered what it will look like if for 4 or 5 days simulation, which is the typical lifetime of atmospheric nitrate. In other words, why picked 48 hours?

Two things are happening during these simulations. The first is near instantaneous isotope fractionation during the photochemistry and the second is isotope mass balance. Similar to the simulations, the real world atmosphere will reflect these two competing processes. At the beginning of the simulations the isotopic change in secondary N compounds ($NO_y$ except $NO_x$) is large because there is no initial concentration of these compounds and their resulting $\delta^{15}N$ is entirely due to the isotope fractionation factors. As the simulation progresses those $NO_y$ compounds that are stable and build up, $HNO_3$ in particular will always approach the $\delta^{15}N$ of the NO emission by mass balance, ie by the end of the five day simulation 99% of the emitted NO has been converted to $HNO_3$ and thus must approximately have the same $\delta^{15}N$ as the NO because is a **closed system**. In the real world the system is open, either due to deposition (wet/dry removal of $HNO_3$) or advective transport. Our 48 hour choice was based on the size of the city and a low average windspeed which would replace the regional air mass at least every 48 hours. A 5 day simulation is imposing a stagnant 5 day air mass over the city, which is unrealistic. The model highlight the importance of atmospheric lifetimes in controlling $\delta^{15}N$ of $NO_y$. At very short lifetimes (post rainy day) the $\delta^{15}N$ should partition strongly, but with long lifetimes the values are controlled by mass balance, in particular long lived reservoirs like $HNO_3$ or PAN. The annual trend is driven by hours of daylight. Something we will explore in detail in subsequent paper.

5. *During the day time, the model result indicates that NO-NO2 isotope exchange is very small compared to the Leighton cycle and the OH reaction. I just wanted to see more data to prove this, i.e., can the authors compare the rate of exchange versus the reaction rates of the Leighton cycle and OH reaction during the daytime?*

The daytime results show $NO_x$ exchange lite is similar to or less than Leighton, but is condition dependent. Both these lifetimes are on the order of 100 sec during peak sunlight and moderate $NO_x$ mixing rations (few 10s ppb). The lifetime of $NO_2$ with respect to $OH + NO_2$ is several hours at peak sunlight. The question is the exchange relative to Leighton which will be a function of $NO_x$ mixing ratio and changing j coefficients. Evaluating this is the purpose of figure 8 (in original). Those show at high NOx exchange is still dominant even during the daytime (8a),

but Leighton becomes dominant at low $NO_x$ during day, and slowly achieve equilibrium at night (8c)

At steady state $-NO_2/dt = NO_2/dt = 1/j_{NO2} = 1/(.001s) = 100$ s

$-dNO2/dt = HNO_3/dt = 1/k[OH] = 1/(6.3*10^{-11})(2.5 \times 10^6) = 6350$ s

6. The last, I suggest to authors to add another case simulation, that in the middle of 5 day simulation, varying the d15N of emitted NO from 0 to, e.g., 10 permil, and see how the isotopes of NOy in the system vary. This will be interesting as in real environment, NOx emitted from different sources and its isotope vary all the time.

   This would be interesting, but we avoid this simulation for two reasons. The first is that the entire objective of the paper is to show potential $\delta^{15}N$ changes in the absence of any $\delta^{15}N$ "sources effect". This objective is laid out on page 4, lines 34-42. Adding source variation would add layers of complexity and add confusion.   Second the emissions in this box model are at a fixed rate (and ratio) and would require reprogramming the model to change on a ½ hour basis (time step) and there is not much current evidence on how these ratio emissions would values change hourly in the real world. This paper we explicitly say that this is only evaluating the photochemical effect on $\delta^{15}N$ . We have new papers in review that address the source effect, and another that will assess source (and mixing) and chemistry combined.

The following are some general comments:

1. P6, line 38-39, this approach has also been mentioned by Bao et al. 2015 (https://doi.org/10.1016/j.gca.2015.07.038) and He et al. 2020 (https://bg.copernicus.org/preprints/bg-2020-120/).

   The quantum approach goes all the way back to Urey, Mayer, and Biegelisen! and is common knowledge in the field

2. Section 2.3.1, for the sensitivity tests in this and other similar sections, it is unclear how many chemistry are involved. For example, in figure 2, was nighttime chemistry involved? If yes, why d15N of HNO3 stays the same at night but when d15N of NO2 is very low?

   We are not fully understanding the reviewer's question.  In every sensitivity test all N reactions are replicated and all a=1 except for one reaction. This is repeated for each N reaction.  For each of those 96 simulations we test if $NO_x$, HONO, or HNO$_3$ $\delta^{15}N$ changes by 1 permil or more. Fig 1 (1) shows a non-sensative reaction (NO3 + NO) and Fig. 2 shows a sensitive reaction (NO$_2$→NO + O). How $\delta^{15}N$  is partitioned the way they do is a complex function initial concentrations, emission rates, simulation length, isotope mass balance…etc.  These are discussed in the subsequent sections.

3. *Section 2.3.1, the second paragraph, the discussion on daytime NO3 and NO2O5, I think this can be made less complicated to just show the mass of these molecules during the daytime, their negligible mass during daytime is the reason of their negligible isotope effects.*

We agree that NO3, $N_2O_5$ is low at night, but this is not the intent of the section. The intent was to show that "sensitive/nonsensative" classification of the reactions was only valid using the $NO_x$, $HNO_3$, HONO as the test molecules. The intent of this section is to show that reactions producing/consuming NO3 and $N_2O_5$ could impact their $\delta^{15}N$ values (ie. NO3 photolysis) but not necessarily the $NO_x$, $HNO_3$, and HONO. This effect would be independent of their concentration. The point being that IF you wanted to model NO3 $\delta^{15}N$ then one would need the more exotic fractionation factors, but not for the main compounds.

4. *Section 3.14, what is eplison-48? If you meant the NO2 + Oh reaction, won't it be e-39?*

Yes, this was a typo!

5. *Section 3.3.5, line 35: .d15N(NO-NO2) = + 20 ‰, but From Figure 15, I didn't see this 20 per mil difference in early nighttime of 1/2 and 6/2;*
6. *Same section, line 37: Shouldn't be Jan. 2 and Jun 2?*
7. *Same section, last sentence: "conditions it requires about 6 hours for NOx to achieve full isotopic equilibrium", I doubt this. First, Walters et al. paper in GRL 2016 actually shows the exchange is fast. In addition, in this simulations, it seems the d15N difference between NO and NO2 reach the maximum by late night, but this could be a result of mass balance, i.e., when almost all NO is converted to NO2, and d15N of NO2 approaches to zero (the starting value), and by mass balance NO will be very negative. It will be much more helpful to understand this if the fraction of NO overnight can be plotted. This issue also exists for all other case simulations. In addition, the model has continued NO emissions with d15N of zero (not a completely closed system), how does this continued add-up of NO affect the isotopes of the system?*

8. *P28, Line 35-36: can you pull out the rate of isotope exchange between NO and NO2, as well as the rate of photolysis and Leighton cycle at daytime and night? it is surprising that at daytime isotope exchange appears to be negligible.*
9. *P29, Line 11-12, what is SI Fig. X? and there is no OH figure in SI.*

*These are all explainable, and will be addressed per comment reply 1.*

---

## Author Response (AR2)

1) whether the model predicts nitrate qualitatively consistent with observations, e.g., in the model, how much daytime nitrate production vs how much nighttime production, and is the relative importance the same as observations?

2) in the Tucson validation case, how is nitrate concentration in the model compared to the observations? the model reproduced the observed d15N very well, indicating nitrate in the city (observations) were mainly from local conversion of NOx to nitrate. It would be simply pull out the model predicted nitrate mass (seasonal pattern if it is difficult to convert the modeled nitrate production to air concentration and compare with the observations directly) and compare with observations. In other words, if the model can't produce the mass right (it is often the case even for a chemical transport model), something strange may happen and need to be clarified.

Thank you for the comment. Some of these concerns were addresses in early versions but were deleted to limit manuscript length, but can be added back with elimination of the case studies. The reviewer is asking two questions that are important and subtlety different. The first is whether the mechanism (model) is accurately capturing photochemistry. The mechanism itself has been reviewed and validated by numerous studies over the past 20 years so we will not address that here. However, did our modification alter the validity of the mechanism? To test this, we ran the 24 test cases studies described in Stockwell et al 1997, without our addition of aerosol reaction or $O_3$ deposition, and compared $O_3$ predicted by RACM and inRACM and they matched identically. Thus our addition of 96 isotopologue reactions is not impacting the oxidation chemistry of RACM. We added the below text and a new Supplementary material figure.

P. 7 text added
*We also tested whether the addition of $^{15}N$ isotopologues had any effect on the RACM's predictions of trace gases over time. Plots of mixing ratios of trace gases such as $HNO_3$ and $O_3$ predicted by RACM versus those $i_NRACM$ run under the same conditions (see Stockwell's 24 simulation tests) yield a slope of 1 with an $R^2 > 0.99$, which expected since the addition of $^{15}N$ compounds is only about 0.3 % of total $NO_x$ and thus should not differ from the RACM predictions.*

The second reviewer comment hints at something often ignored when using 0-D box models and and that is whether the model can predict real world trace gas concentrations? Since RACM and other mechanisms are usually validated by chamber studies where emission, deposition and dilution are often not relevant (chemistry in an inert box) they can be a challenge to use when comparing to real world trace gas concentrations. This is important since the $NO_y$ $\delta^{15}N$ is reflecting different isotope fractionations in different oxidation pathways. We spent a spend time examining this at the beginning of the study when simulations would produce $O_3$ mixing ratios that were unreasonable based on observations, which in turn amplified oxidation state of the model. We address this with the added text and a new figure in Supplement, which is also used to assess the accuracy of the Tucson simulations.

P 16.

*2.4.4 Addition of $O_3$ deposition to $i_N$RACM*

*Photochemical mechanisms such as RACM are validated by comparing model predictions with observed trace gas concentration evolution in chambers studies, which has its limitations. For example, Stockwell et. al. compared RACM, RADM2, and SPARC mechanisms ability to predict trace gases concentrations (e.g. $O_3$, $NO_2$, toluene) with those observed in chamber experiments (see Stockwell et. al., Fig 3-9) and achieve good agreement between the model and experiments. These experiment-model comparisons essentially validate the rate constant assumptions in the chemical mechanism. Box models are, however, limited in their ability to predict real world concentrations because many do not account for pollutant deposition (dry or wet) since these are handled when the mechanism is incorporated into 1, 2 and 3D transport models. Similarly, dilution by of trace gases due to vertical (or horizontal) transport is typically not incorporated into 0-box models. This can lead to the buildup (or depletion) of key oxidants, particularly $O_3$ (see Fig. 6 in Stockwell et.al.). This in turn will significantly alter $NO_x$ oxidation pathways, and since the $\delta^{15}N$ in $i_N$RACM is effectively a function on changing oxidation pathways, this would impact $i_N$RACM ability to accurately predict the observations of $\delta^{15}N$ in the real world. In order to eliminate this bias, we added a $O_3$ deposition reaction and adjusted the rate until $O_3$ mixing ratios were in line with typical suburban mixing ratios (20-30 ppb) and exhibited a typical $O_3$ diurnal mixing ratio variation, low nighttime/high midday, that are observed in most environments (Fig S2). This results in simulated daytime maximum OH concentrations on the order of ~ 8 x $10^6$ molecules $cm^{-3}$ and daytime average of ~ 2 x $10^6$ molecules $cm^{-3}$ (Fig S2) that are typical of overserved concentrations in urban and suburban environments (see refs. in the review by Monks, 2005). This gives us confidence that $i_N$RACM is accurately capturing boundary layer photochemistry and can be used to predict $\delta^{15}N$ in $NO_y$ compounds.*

As to the reviewers question about day/night $HNO_3$ production we added the following.

P. 27

*Analysis of hourly $HNO_3$ production (in June) revealed that ~80% of $HNO_3$ is produced in the daytime, mainly by the $NO_2$ + OH reaction and 20% is produced during the night ($N_2O_5$ heterogenous hydrolysis). The model reproduces $O_3$ and $NO_x$ concentrations rather accurately (Fig S2) but $HNO_3$ concentrations that are about 10 times the PM $NO_3^-$ concentration. This is not surprising because the 0-D models do not account for $HNO_3$ deposition, its dilution as it mixes into to the top of the boundary layer, or partitioning between aerosol and the gas phase. Indeed, seasonal differences in boundary layer height alone can dilute by a factor of 5 or higher [Riha, 2013].*